# Stochastic Mirror Descent on Overparameterized Nonlinear Models

## Abstract

Most modern learning problems are highly *overparameterized*, meaning that the model has many more parameters than the number of training data points, and as a result, the training loss may have infinitely many global minima (in fact, a manifold of parameter vectors that perfectly interpolates the training data). Therefore, it is important to understand which interpolating solutions we converge to, how they depend on the initialization point and the learning algorithm, and whether they lead to different generalization performances. In this paper, we study these questions for the family of stochastic mirror descent (SMD) algorithms, of which the popular stochastic gradient descent (SGD) is a special case. Recently it has been shown that, for overparameterized *linear* models, SMD converges to the global minimum that is "closest" (in terms of the Bregman divergence of the mirror used) to the initialization point, a phenomenon referred to as *implicit regularization*. Our contributions in this paper are both theoretical and experimental. On the theory side, we show that in the overparameterized *nonlinear* setting, if the initialization is *close enough* to the manifold of global optima, SMD with sufficiently small step size converges to a global minimum that is approximately the closest global minimum in Bregman divergence, thus attaining *approximate implicit regularization*. For highly overparametrized models, this closeness comes for free: the manifold of global optima is so high dimensional that with high probability an arbitrarily chosen initialization will be close to the manifold. On the experimental side, our extensive experiments on the MNIST and CIFAR-10 datasets, using various initializations, various mirror descents, and various Bregman divergences, consistently confirms that this phenomenon indeed happens in deep learning: SMD converges to the closest global optimum to the initialization point in the Bregman divergence of the mirror used. Our experiments further indicate that there is a clear difference in the generalization performance of the solutions obtained from different SMD algorithms. Experimenting on the CIFAR-10 dataset with different regularizers, $\ell_1$ to encourage sparsity, $\ell_2$ (yielding SGD) to encourage small Euclidean norm, and $\ell_{10}$ to discourage large components in the parameter vector, consistently and definitively shows that, for small initialization vectors, $\ell_{10}$-SMD has better generalization performance than SGD, which in turn has better generalization performance than $\ell_1$-SMD. This surprising, and perhaps counter-intuitive, result strongly suggests the importance of a comprehensive study of the role of regularization, and the choice of the best regularizer, to improve the generalization performance of deep networks.

## 1 Introduction

Deep learning has demonstrably enjoyed a great deal of success in a wide variety of tasks (Amodei et al., 2016; Graves et al., 2013; Krizhevsky et al., 2012; Mnih et al., 2015; Silver et al., 2016; Wu et al., 2016; LeCun et al., 2015). Despite its tremendous success, the reasons behind the good performance of these methods on unseen data is not fully understood (and, arguably, remains somewhat of a mystery). While the special deep architecture of these models seems to be important to the success of deep learning, the architecture is only part of the story, and it has been now widely recognized that the optimization algorithms used to train these models, typically stochastic gradient descent (SGD) and its variants, play a key role in learning parameters that generalize well.

Since these deep models are *highly overparameterized*, they have a lot of capacity, and can fit to virtually any (even random) set of data points (Zhang et al., 2016). In other words, these highly overparameterized models can "interpolate" the training data, so much so that this regime has been called the "interpolating regime" (Ma et al., 2018b). In fact, on a given dataset, the loss function typically has (infinitely) many *global minima*, which however can have drastically different generalization properties (many of them perform very poorly on the test set). Which minimum among all the possible minima we converge to in practice is determined by the initialization and the optimization algorithm that we use for training the model.

Since the loss functions of deep neural networks are non-convex—sometimes even non-smooth—in theory, one may expect the optimization algorithms to get stuck in local minima or saddle points. In practice, however, such simple stochastic descent algorithms almost always reach *zero training error*, i.e., a *global minimum* of the training loss (Zhang et al., 2016; Lee et al., 2016). More remarkably, even in the absence of any explicit regularization, dropout, or early stopping (Zhang et al., 2016), the global minima obtained by these algorithms seem to generalize quite well (contrary to some other global minima). It has been also observed that even among different optimization algorithms, i.e., SGD and its variants, there is a discrepancy in the solutions achieved by different algorithms and how they generalize (Wilson et al., 2017). Therefore, it is important to ask

> *Which global minima do these algorithms converge to? And what properties do they have?*

In this paper, we answer this question for the SGD algorithm, and more generally, for the family of stochastic mirror descent (SMD) algorithms, which includes SGD as a special case. For any choice of potential function, there is a corresponding mirror descent algorithm. We show that, for overparameterized nonlinear models, if one initializes close enough to the manifold of parameter vectors that interpolates the data, then the SMD algorithm for any particular potential converges to a global minimum that is approximately ***the closest one to the initialization, in Bregman divergence corresponding to the potential.*** In highly overparameterized models, this closeness of the initialization comes for free, something that is occasionally referred to as "the blessing of dimensionality." For the special case of SGD, this means that it converges to a global minimum which is approximately the closest one to the initialization in the usual Euclidean sense.

We perform extensive systematic experiments with various initial points and various mirror descent algorithms for the MNIST and CIFAR-10 datasets using standard off-the-shelf deep neural network architectures for these datasets with standard random initialization, and we measure all the resulting pairwise Bregman divergences. We found that every single result is exactly consistent with the above theory. Indeed, in all our experiments, ***the global minimum achieved by any particular mirror descent algorithm is the closest, among all other global minima obtained by other mirrors and other initializations, to its initialization in the corresponding Bregman divergence.*** In particular, the global minimum obtained by SGD from any particular initialization is closest to the initialization in Euclidean sense, both among the global minima obtained by different mirrors and among the global minima obtained by different initializations.

This result, proven theoretically and backed up by extensive experiments, further implies that, even in the absence of any explicit regularization, these algorithms perform an *implicit regularization*. In particular, it implies that, when initialized around zero, SGD acts as an $\ell_2$ regularizer. Similarly, by choosing other mirrors, one obtains different forms of implicit regularization (such as $\ell_1$ or $\ell_\infty$), which may have different performances on test data. This raises the question

> *How well do different mirrors perform in practice?*

Perhaps, one might expect an $\ell_1$ regularizer to perform better, due to the fact that it promotes sparsity, and "pruning" in neural networks is believed to be helpful for generalization. On the other hand, one may expect SGD ($\ell_2$ regularizer) to work best among different mirrors, because typical architectures have been tuned for and tailored to SGD. We run experiments with four different mirror descents, i.e., $\ell_1$ (sparse), $\ell_2$ (SGD), $\ell_3$, and $\ell_{10}$ (as a surrogate for $\ell_\infty$), on a standard off-the-shelf deep neural network architecture for CIFAR-10, namely ResNet-18. ***Somewhat counter-intuitively, our results for test errors of different mirrors consistently and definitively show that the $\ell_{10}$ regularizer performs better than the other mirrors including SGD, while $\ell_1$ consistently performs worse.*** This flies in the face of the conventional wisdom that sparser weights (which are obtained by an $\ell_1$ regularizer) generalize better, and suggests that $\ell_\infty$, which roughly speaking penalizes all the weights uniformly, may be a better regularizer for deep neural networks.

## 1.1 RELATED WORK

There have been many efforts in the past few years to study deep learning from an optimization perspective, e.g., (Achille & Soatto, 2017; Chaudhari & Soatto, 2018; Shwartz-Ziv & Tishby, 2017; Allen-Zhu et al., 2019; Oymak & Soltanolkotabi, 2019; Azizan & Hassibi, 2019; Ma et al., 2018b; Du et al., 2018; Li & Liang, 2018; Cao & Gu, 2019). While it is not possible to review all the contributions here, we comment on the ones that are most closely related to ours and highlight the distinctions between our results and those.

Many recent papers have studied the convergence of the (S)GD algorithm in the so-called "over-parameterized" setting (or "interpolating" regime), which is common in deep learning (Oymak & Soltanolkotabi, 2019; Allen-Zhu et al., 2019; Soltanolkotabi et al., 2017; Ma et al., 2018b). All these works, similar to ours, assume that the initialization is close to the solution space (of global minima), which is a reasonable assumption in highly overparameterized models, as we argue in Section A.4 of the supplementary material. However, our results are more general because they extend to SMD.

Furthermore, even for the case of SGD, our results are stronger than those in the literature, in the sense that not only do we show convergence to a global minimum, but we also show that the weight vector we converge to, $w_\infty$, say, is close to the interpolating weight vector closest to the initialization, $w^*$, say. Denoting the initialization by $w_0$, Oymak & Soltanolkotabi (2019) showed that for SGD, $\|w_\infty - w_0\|$ is bounded by a constant factor of $\|w^* - w_0\|$. Our Theorem 4 shows the much stronger statement that $\|w_\infty - w_0\| = \|w^* - w_0\| + o(\|w^* - w_0\|)$. We further show that $w_\infty$ and $w^*$ are very close to one another, viz. $\|w_\infty - w^*\|^2 = o(\|w^* - w_0\|)$, something that could not be inferred from the previous work.

There exist a number of results that characterize the implicit regularization properties of different algorithms in different contexts (Neyshabur et al., 2017; Ma et al., 2018a; Gunasekar et al., 2017; 2018a; Soudry et al., 2017; Gunasekar et al., 2018b; Azizan & Hassibi, 2019; Mianjy et al., 2018). The closest ones to our results, since they concern mirror descent, are the works of (Gunasekar et al., 2018a; Azizan & Hassibi, 2019). The authors in (Gunasekar et al., 2018a) consider *linear* overparameterized models, and show that *if* SMD happens to converge to a global minimum, then that global minimum will be the one that is closest in Bregman divergence to the initialization, a result they obtain by examining the KKT conditions. However, they do not provide any conditions for convergence and whether SMD converges with a fixed step size or not. Azizan & Hassibi (2019) also study linear models, but derive conditions on the step size for which SMD converges to the aforementioned global minimum. Our results extend the aforementioned to *nonlinear* overparametrized models, and show that, for small enough *fixed* step size, and for initializations close enough to the space of interpolating solutions, SMD converges to a global minimum, something which had not been shown in any of the previous work. Assuming every data point is revisited often enough, the convergence we establish is *deterministic*. Finally, we show that the solution we converge to exhibits approximate implicit regularization, something that was not not known for nonlinear models.

The remander of the paper is organized as follows. In Section 2, we review the family of mirror descent algorithms and briefly revisit the linear overparameterized case. Section 3 provides our main theoretical results, which are (1) convergence of SMD to a *global* minimum, and (2) proximity of the obtained global minimum to the closest one from the initialization in Bregman divergence. Our proofs are remarkably simple and are based on a powerful *fundamental identity* that holds for all SMD algorithms in a general setting. In Section 4, we provide our experimental results, which consists of two parts, (1) testing the theoretical claims about the distances for different mirrors and different initializations, and (2) assessing the generalization properties of different mirrors. The proofs of the theoretical results and more details on the experiments are relegated to the supplementary material.

## 2 BACKGROUND AND WARM-UP

Let us denote the training dataset by $\{(x_i, y_i) : i = 1, \ldots, n\}$, where $x_i \in \mathbb{R}^d$ are the inputs, and $y_i \in \mathbb{R}$ are the labels. The model (which can be, e.g., linear, a deep neural network, etc.) is defined by the general function $f(x_i, w) = f_i(w)$ with some parameter vector $w \in \mathbb{R}^p$. Since typical deep models have a lot of capacity and are highly overparameterized, we are particularly interested in the overparameterized (or so-called interpolating) regime, where $p > n$ (often $p \gg n$). In this case,

there are many parameter vectors $w$ that are consistent with the training data points. We denote the set of these parameter vectors by

$$\mathcal{W} = \{w \in \mathbb{R}^p \mid f(x_i, w) = y_i, i = 1, \ldots, n\} \tag{1}$$

This a high-dimensional set (e.g. a $(p - n)$-dimensional manifold) in $\mathbb{R}^p$ and depends only on the training data $\{(x_i, y_i) : i = 1, \ldots, n\}$ and the model $f(\cdot, \cdot)$.

The total loss on the training set (empirical risk) can be expressed as $L(w) = \sum_{i=1}^{n} L_i(w)$, where $L_i(\cdot) = \ell(y_i, f(x_i, w))$ is the loss on the individual data point $i$, and $\ell(\cdot, \cdot)$ is a differentiable non-negative function, with the property that $\ell(y_i, f(x_i, w)) = 0$ iff $y_i = f(x_i, w)$. Often $\ell(y_i, f(x_i, w)) = \ell(y_i - f(x_i, w))$, with $\ell(\cdot)$ convex and having a global minimum at zero (such as square loss, Huber loss, etc.). In this case, $L(w) = \sum_{i=1}^{n} \ell(y_i - f(x_i, w))$.

$\mathcal{W}$ is the set of global minima, and every parameter vector $w$ in $\mathcal{W}$ renders the loss on each data point zero, i.e., $L_i(w) = 0 \; \forall i$. The loss function is often attempted to be minimized by stochastic gradient descent (SGD):

$$w_i = w_{i-1} - \eta \nabla L_i(w_{i-1}), \quad i \geq 1 \tag{2}$$

assuming the data is indexed randomly. We use one index $i$ for both the loss and the parameter vector at step $i$ (for $i > n$, one can cycle through the data, or select them at random, etc.).

## 2.1 STOCHASTIC MIRROR DESCENT

Mirror descent, first introduced by Nemirovski & Yudin (1983), is one of the most widely used families of algorithms for optimization (Beck & Teboulle, 2003; Cesa-Bianchi et al., 2012; Zhou et al., 2017), which includes the popular gradient descent as a special case. Consider a strictly convex differentiable function $\psi(\cdot)$, called the *potential function*. Then updates for stochastic mirror descent (SMD) are defined as

$$\nabla \psi(w_i) = \nabla \psi(w_{i-1}) - \eta \nabla L_i(w_{i-1}). \tag{3}$$

Note that, due to the strict convexity of $\psi(\cdot)$, the gradient $\nabla \psi(\cdot)$ defines an invertible map, so the recursion in (3) yields a unique $w_i$ at each iteration, and thus is a well-defined update, i.e., $w_i = \nabla \psi^{-1} \left( \nabla \psi(w_{i-1}) - \eta \nabla L_i(w_{i-1}) \right)$. Compared to classical SGD, rather than update the weight vector along the direction of the negative gradient, the update is done in the "mirrored" domain determined by the invertible transformation $\nabla \psi(\cdot)$. Mirror descent was originally conceived to exploit the geometrical structure of the problem by choosing an appropriate potential. Note that SMD reduces to SGD when $\psi(w) = \frac{1}{2}\|w\|^2$, since the gradient $\nabla \psi(\cdot)$ is simply the identity map.

Alternatively, the update rule (3) can be expressed as

$$w_i = \arg\min_{w} \; \eta w^T \nabla L_i(w_{i-1}) + D_\psi(w, w_{i-1}), \tag{4}$$

where

$$D_\psi(w, w_{i-1}) := \psi(w) - \psi(w_{i-1}) - \nabla \psi(w_{i-1})^T (w - w_{i-1}) \tag{5}$$

is the Bregman divergence with respect to the potential function $\psi(\cdot)$. Note that $D_\psi(\cdot, \cdot)$ is non-negative, convex in its first argument, and that, due to strict convexity, $D_\psi(w, w') = 0$ iff $w = w'$.

Different choices of the potential function $\psi(\cdot)$ yield different optimization algorithms, which will potentially have different implicit biases. A few examples follow.

**Gradient Descent.** For the potential function $\psi(w) = \frac{1}{2}\|w\|^2$, the Bregman divergence is $D_\psi(w, w') = \frac{1}{2}\|w - w'\|^2$, and the update rule reduces to that of SGD.

**Exponentiated Gradient Descent.** For $\psi(w) = \sum_j w_j \log w_j$, the Bregman divergence becomes the unnormalized relative entropy (Kullback-Leibler divergence) $D_\psi(w, w') = \sum_j w_j \log \frac{w_j}{w'_j} - \sum_j w_j + \sum_j w'_j$, which corresponds to the exponentiated gradient descent (aka the exponential weights) algorithm (Kivinen & Warmuth, 1997).

$p$**-norms Algorithm.** For any $q$-norm squared potential function $\psi(w) = \frac{1}{2}\|w\|_q^2$, with $\frac{1}{p} + \frac{1}{q} = 1$, the algorithm will reduce to the so-called $p$-norms algorithm (Grove et al., 2001; Gentile, 2003).

**Sparse Mirror Descent.** For $\psi(w) = \|w\|_{1+\epsilon}^{1+\epsilon}$, the algorithm reduces to sparse mirror descent, which is used in compressed sensing (Azizan & Hassibi, 2019).

## 2.2 OVERPARAMETERIZED LINEAR MODELS

Overparameterized (or underdetermined) linear models have been recently studied in many papers due to their simplicity, and the fact that there are interesting insights than one can obtain from them. In this case, the model is $f(x_i, w) = x_i^T w$, the set of global minima is $\mathcal{W} = \{w \mid y_i = x_i^T w, \; i = 1, \ldots, n\}$, and the loss is $L_i(w) = \ell(y_i - x_i^T w)$. The following result characterizes the solution that SMD converges to (Azizan & Hassibi, 2019; Gunasekar et al., 2018a).

**Proposition 1.** *Consider a linear overparameterized model. For sufficiently small step size, i.e., for any $\eta > 0$ for which $\psi(\cdot) - \eta L_i(\cdot)$ is convex, and for any initialization $w_0$, the SMD iterates converge to*

$$w_\infty = \arg\min_{w \in \mathcal{W}} D_\psi(w, w_0).$$

Note that the step size condition, i.e., the convexity of $\psi(\cdot) - \eta L_i(\cdot)$, depends on both the loss and the potential function. For the case of SGD, $\psi(w) = \frac{1}{2}\|w\|^2$, and $\ell(y_i - x_i^T w) = \frac{1}{2}(y_i - x_i^T w)^2$, so the condition reduces to the well-known $\eta \leq \frac{1}{\|x_i\|^2}$. In this case, $D_\psi(w, w_0)$ is simply $\frac{1}{2}\|w - w_0\|^2$.

**Corollary 2.** *In particular, for the initialization $w_0 = \arg\min_{w \in \mathbb{R}^p} \psi(w)$, under the conditions of Proposition 1, the SMD iterates converge to*

$$w_\infty = \arg\min_{w \in \mathcal{W}} \psi(w). \tag{6}$$

This means that running SMD for a linear model with the aforementioned $w_0$, without any explicit regularization, results in a solution that has the smallest potential $\psi(\cdot)$ among all solutions, i.e., SMD implicitly regularizes the solution with $\psi(\cdot)$. In particular, this means that SGD initialized around zero acts an an $\ell_2$-norm regularizer. In this paper, we show that these results continue to hold for highly overparameterized nonlinear models in an approximate sense.

## 3 THEORETICAL RESULTS

In this section, we provide our main theoretical results. In particular, we show that for highly over-parameterized models (1) SMD converges to a global minimum, (2) the global minimum obtained by SMD is approximately the closest one to the initialization in Bregman divergence corresponding to the potential.

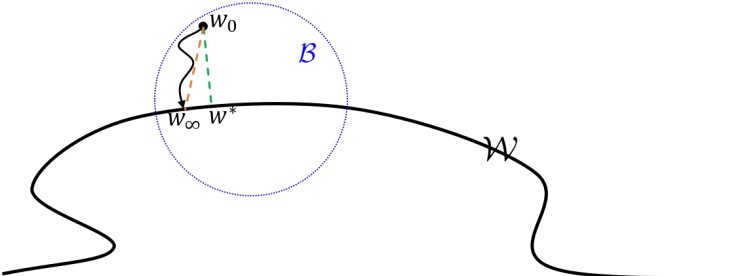

Figure 1: An illustration of the parameter space. $\mathcal{W}$ represents the set of global minima, $w_0$ is the initialization, $\mathcal{B}$ is the local neighborhood, $w^*$ and the closest global minimum to $w_0$ (in Bregman divergence), and $w_\infty$ is the minimum that SMD converges to.

### 3.1 CONVERGENCE OF STOCHASTIC MIRROR DESCENT

Let us define

$$D_{L_i}(w, w') := L_i(w) - L_i(w') - \nabla L_i(w')^T(w - w'), \tag{7}$$

which is defined in a similar way to a Bregman divergence for the loss function. The difference though is that, unlike the potential function of the Bregman divergence, the loss function $L_i(\cdot) = \ell(y_i - f(x_i, \cdot))$ need not be convex (even when $\ell(\cdot)$ is) due to the nonlinearity of $f(\cdot, \cdot)$.

It has been argued in several recent papers that in highly overparameterized neural networks, any random initialization $w_0$ is close to $\mathcal{W}$, with high probability (Li & Liang, 2018; Du et al., 2018; Azizan & Hassibi, 2019; Allen-Zhu et al., 2019; Cao & Gu, 2019) (see also the discussion in Section A.4 of the supplementary material). Therefore, it is reasonable to make the following assumption about the initialization.

**Assumption 1.** *Denote the initial point by $w_0$. There exists $w \in \mathcal{W}$ and a region $\mathcal{B} = \{w' \in \mathbb{R}^p \mid D_\psi(w, w') \leq \epsilon\}$ containing $w_0$, such that $D_{L_i}(w, w') \geq 0, i = 1, \ldots, n$, for all $w' \in \mathcal{B}$.*

It is important to understand what this assumption means. Since $L_i(\cdot)$ is not necessarily convex, it is certainly not the case that $D_{L_i}(w, w') \geq 0$ for all $w'$. However, since $w$ is a minimizer of $L_i(\cdot)$, there will be a neighborhood around it such that for all $w'$ in this neighborhood $D_{L_i}(w, w') \geq 0$ (see Fig. 2 for an illustration). What we are requiring is that the initialization $w_0$ be inside the intersection of all such neighborhoods for $i = 1, 2, \ldots, n$. In other words, we require a $w_0$ close enough to $\mathcal{W}$.

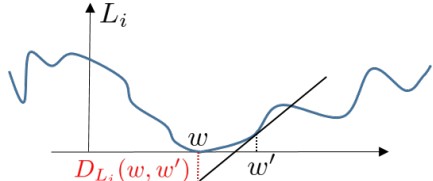

Figure 2: An illustration of $D_{L_i}(w, w') \geq 0$ in a local region in Assumption 1.

Our second assumption states that in this local region, the first and second derivatives of the model are bounded.

**Assumption 2.** *Consider the region $\mathcal{B}$ in Assumption 1. $f_i(\cdot)$ have bounded gradient and Hessian on the convex hull of $\mathcal{B}$, i.e., $\|\nabla f_i(w')\| \leq \gamma$, and $\alpha \leq \lambda_{\min}(H_{f_i}(w')) \leq \lambda_{\max}(H_{f_i}(w')) \leq \beta, i = 1, \ldots, n$, for all $w' \in \mathrm{conv}\ \mathcal{B}$.*

This is again a mild assumption, which is assumed in other related work such as (Oymak & Soltanolkotabi, 2019) as well. Note that we do *not* require $\alpha$ to be positive (just its boundedness). The following theorem states that under Assumption 1, SMD converges to a global minimum.

**Theorem 3.** *Consider the set of interpolating parameters $\mathcal{W} = \{w \in \mathbb{R}^p \mid f(x_i, w) = y_i, i = 1, \ldots, n\}$, and the SMD iterates given in (3), where every data point is revisited after some steps. Under Assumption 1, for sufficiently small step size, i.e., for any $\eta > 0$ for which $\psi(\cdot) - \eta L_i(\cdot)$ is strictly convex on $\mathcal{B}$ for all $i$, the following holds.*

1. *All the iterates $\{w_i\}$ remain in $\mathcal{B}$.*

2. *The iterates converge (to $w_\infty$).*

3. *$w_\infty \in \mathcal{W}$.*

Note that, while convergence (to some point) with decaying step size is almost trivial, this result establishes convergence to the solution set with a *fixed* step size. Furthermore, the convergence is *deterministic*, and is not in expectation or with high probability. For example, this result also applies to the case where we cycle through the data deterministically.

We should also remark that the choice of distance in the definition of the "ball" $\mathcal{B}$ was important to be the Bregman divergence with respect to $\psi(\cdot)$ and in that particular order. In fact, one cannot guarantee that the SMD iterates get closer to an interpolating $w$ at every step in the usual Euclidean sense. However, once can establish that it gets closer in $D_\psi(w, \cdot)$. Finally, it is important to note that we need the step size to be small enough to guarantee the strict convexity of $\psi(\cdot) - \eta L_i(\cdot)$ in $\mathcal{B}$, not globally.

Denote the global minimum that is closest to the initialization in Bregman divergence by $w^*$, i.e.,

$$w^* = \arg\min_{w \in \mathcal{W}} D_\psi(w, w_0). \tag{8}$$

Recall that in the linear case, this was what SMD converges to. We show that in the nonlinear case, under Assumptions 1 and 2, SMD converges to a point $w_\infty$ which is "very close" to $w^*$.

**Theorem 4.** *Define $w^* = \arg\min_{w \in \mathcal{W}} D_\psi(w, w_0)$. Under the conditions of Theorem 3, and Assumption 2, the following holds*

1. $D_\psi(w_\infty, w_0) = D_\psi(w^*, w_0) + o(\epsilon)$

2. $D_\psi(w^*, w_\infty) = o(\epsilon)$

In other words, if we start with an initialization that is $O(\epsilon)$ away from $\mathcal{W}$ (in Bregman divergence), we converge to a point $w_\infty \in \mathcal{W}$ that is $o(\epsilon)$ away from $w^*$. The Bregman divergence of this point is $o(\epsilon)$ from the minimum value it can take.

**Corollary 5.** *For the initialization $w_0 = \arg\min_{w \in \mathbb{R}^p} \psi(w)$, under the conditions of Theorem 4, $w^* = \arg\min_{w \in \mathcal{W}} \psi(w)$ and the following holds.*

1. $\psi(w_\infty) = \psi(w^*) + o(\epsilon)$

2. $D_\psi(w^*, w_\infty) = o(\epsilon)$

### 3.2 Proof Technique: Fundamental Identity of SMD

The main tool used for the proofs is a fundamental identity that holds for SMD.

**Lemma 6.** *For any model $f(\cdot, \cdot)$, any differentiable loss $\ell(\cdot)$, any parameter $w \in \mathcal{W}$, and any step size $\eta > 0$, the following relation holds for the SMD iterates $\{w_i\}$*

$$D_\psi(w, w_{i-1}) = D_\psi(w, w_i) + D_{\psi - \eta L_i}(w_i, w_{i-1}) + \eta L_i(w_i) + \eta D_{L_i}(w, w_{i-1}), \quad (9)$$

*for all $i \geq 1$.*

This identity allows one to prove the results in a remarkably simple and direct way. Due to space limitations, the proofs are relegated to the supplementary material.

The ideas behind this identity are related to $H_\infty$ estimation theory (Hassibi et al., 1999; Simon, 2006), which was originally developed in the 1990's in the context of robust control theory. In fact, it has connections to the minimax optimality of SGD, which was shown by (Hassibi et al., 1994) for linear models, and recently extended to nonlinear models and general mirrors by (Azizan & Hassibi, 2019).

## 4 Experimental Results

In this section, we provide our experimental results, which consist of two main parts. In the first part, we evaluate the theoretical claims by running systematic experiments for different initializations and different mirrors, and evaluating the distances between the global minima achieved and the initializations, in different Bregman divergences. In the second part, we assess the generalization error of different mirrors, which correspond to different regularizers, in order to understand which regularizer performs better.

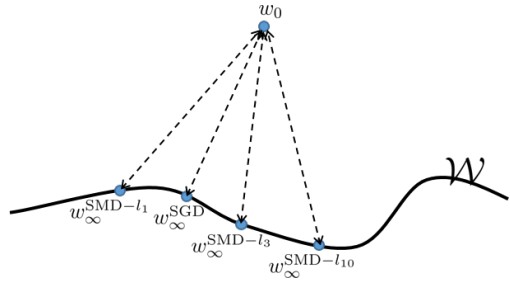

Figure 3: An illustration of the experiments in Table 1

|  | SMD 1-norm | SMD 2-norm (SGD) | SMD 3-norm | SMD 10-norm |
|---|---|---|---|---|
| 1-norm BD | 141 | $9.19 \times 10^3$ | $4.1 \times 10^4$ | $2.34 \times 10^5$ |
| 2-norm BD | $3.15 \times 10^3$ | 562 | $1.24 \times 10^3$ | $6.89 \times 10^3$ |
| 3-norm BD | $4.31 \times 10^4$ | 107 | 53.5 | $1.85 \times 10^2$ |
| 10-norm BD | $6.83 \times 10^{13}$ | 972 | $7.91 \times 10^{-5}$ | $2.72 \times 10^{-8}$ |

Table 1: Fixed Initialization. Distances from final points (global minima) obtained by different algorithms (columns) from the same initialization (Fig. 3), measured in different Bregman divergences (rows). **First Row**: The closest point to $w_0$ in $\ell_1$ Bregman divergence, among the four final points, is exactly the one obtained by SMD with 1-norm potential. **Second Row**: The closest point to $w_0$ in $\ell_2$ Bregman divergence (Euclidean distance), among the four final points, is exactly the one obtained by SGD. **Third Row**: The closest point to $w_0$ in $\ell_3$ Bregman divergence, among the four final points, is exactly the one obtained by SMD with 3-norm potential. **Fourth Row**: The closest point to $w_0$ in $\ell_{10}$ Bregman divergence, among the four final points, is exactly the one obtained by SMD with 10-norm potential.

### 4.1 Do SMDs Converge to the Closest Point in Bregman Divergence?

While accessing all the points on $\mathcal{W}$ and finding the closest one is impossible, we design systematic experiments to test this claim. We run experiments on some standard deep learning problems, namely, a standard CNN on MNIST (LeCun et al., 1998) and the ResNet-18 (He et al., 2016) on CIFAR-10 (Krizhevsky & Hinton, 2009). We train the models from different initializations, and with different mirror descents from each particular initialization, until we reach $100\%$ training accuracy, i.e., a point on $\mathcal{W}$. We randomly initialize the parameters of the networks around zero. We choose 6 independent initializations for the CNN, and 8 for ResNet-18, and for each initialization, we run different SMD algorithms with the following four potential functions: (a) $\ell_1$ norm, (b) $\ell_2$ norm (which is SGD), (c) $\ell_3$ norm, (d) $\ell_{10}$ norm (as a surrogate for $\ell_\infty$). See Supplementary Material B for more details on the experiments.

|  | Final 1 | Final 2 | Final 3 | Final 4 | Final 5 | Final 6 | Final 7 | Final 8 |
|---|---|---|---|---|---|---|---|---|
| Initial 1 | $6 \times 10^2$ | $2.9 \times 10^3$ | $2.8 \times 10^3$ | $2.8 \times 10^3$ | $2.8 \times 10^3$ | $2.8 \times 10^3$ | $2.8 \times 10^3$ | $2.8 \times 10^3$ |
| Initial 2 | $2.8 \times 10^3$ | $6.1 \times 10^2$ | $2.8 \times 10^3$ | $2.8 \times 10^3$ | $2.8 \times 10^3$ | $2.8 \times 10^3$ | $2.8 \times 10^3$ | $2.8 \times 10^3$ |
| Initial 3 | $2.8 \times 10^3$ | $2.9 \times 10^3$ | $5.6 \times 10^2$ | $2.8 \times 10^3$ | $2.8 \times 10^3$ | $2.8 \times 10^3$ | $2.8 \times 10^3$ | $2.8 \times 10^3$ |
| Initial 4 | $2.8 \times 10^3$ | $2.9 \times 10^3$ | $2.8 \times 10^3$ | $5.9 \times 10^2$ | $2.8 \times 10^3$ | $2.8 \times 10^3$ | $2.8 \times 10^3$ | $2.8 \times 10^3$ |
| Initial 5 | $2.8 \times 10^3$ | $2.9 \times 10^3$ | $2.8 \times 10^3$ | $2.8 \times 10^3$ | $5.7 \times 10^2$ | $2.8 \times 10^3$ | $2.8 \times 10^3$ | $2.8 \times 10^3$ |
| Initial 6 | $2.8 \times 10^3$ | $2.9 \times 10^3$ | $2.8 \times 10^3$ | $2.8 \times 10^3$ | $2.8 \times 10^3$ | $5.6 \times 10^2$ | $2.8 \times 10^3$ | $2.8 \times 10^3$ |
| Initial 7 | $2.8 \times 10^3$ | $2.9 \times 10^3$ | $2.8 \times 10^3$ | $2.8 \times 10^3$ | $2.8 \times 10^3$ | $2.8 \times 10^3$ | $6 \times 10^2$ | $2.8 \times 10^3$ |
| Initial 8 | $2.8 \times 10^3$ | $2.9 \times 10^3$ | $2.8 \times 10^3$ | $2.8 \times 10^3$ | $2.8 \times 10^3$ | $2.8 \times 10^3$ | $2.8 \times 10^3$ | $5.8 \times 10^2$ |

Table 2: Fixed Mirror: SGD. Pairwise distances between different initial points and the final points obtained from them by SGD (Fig. 4). **Row i**: The closest final point to the initial point $i$, among all the eight final points, is exactly the one obtained by the algorithm from initialization $i$.

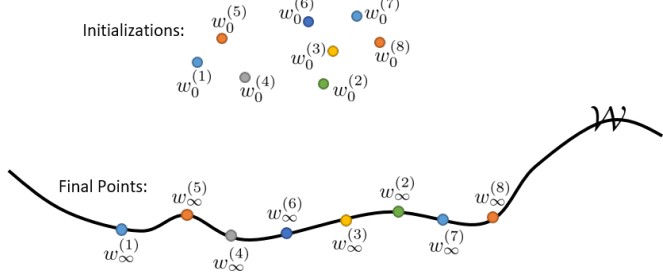

Figure 4: An illustration of the experiments in Table 2

We measure the distances between the initializations and the global minima obtained from different mirrors and different initializations, in different Bregman divergences. Table 1, and Table 2 show some examples among different mirrors and different initializations, respectively. Fig. 5 shows the

distances between a particular initial point and all the final points obtained from different initializations and different mirrors (the distances are often orders of magnitude different, so we show them in logarithmic scale). The global minimum achieved by any mirror from any initialization is the closest in the correct Bregman divergence, among all mirrors, among all initializations, and among both. This trend is very consistent among all our experiments, which can be found in Supplementary Material B.

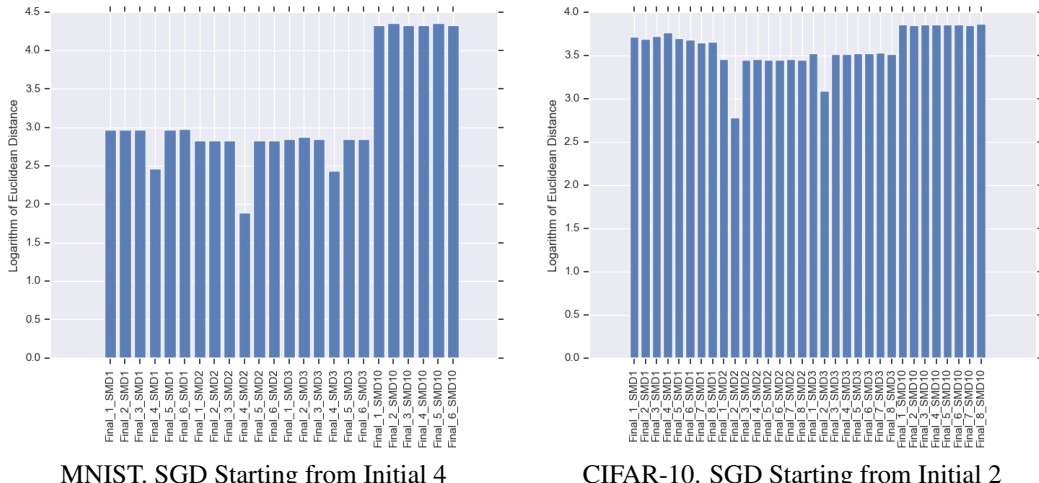

MNIST. SGD Starting from Initial 4          CIFAR-10. SGD Starting from Initial 2

Figure 5: Distances between a particular initial point and all the final points obtained by both different initializations and different mirrors. The smallest distance, among all initializations and all mirrors, corresponds exactly to the final point obtained from that initial point by SGD. This trend is observed consistently for all other mirror descents and all initializations (see the results in Tables 8 and 9 in the appendix).

## 4.2 DISTRIBUTION OF THE WEIGHTS OF THE NETWORK

One may be curious to see how the final weights obtained by these different mirrors look like, and whether, for example, mirror descent corresponding to the $\ell_1$-norm potential induces sparsity. Fig. 6 shows the histogram of the absolute value of the weights for different SMDs, when initialized by the *same* set of close to zero weights. The histograms of the final weights look substantially different and, since they all started from the same initial weights, this difference is fully attributable to the different mirrors used. The histogram of the $\ell_1$-SMD has more weights at and close to zero, which confirms that it induces sparsity. The histogram of the $\ell_2$-SMD (SGD) looks almost perfectly Gaussian, whereas the $\ell_3$ and $\ell_{10}$ histograms are shifted to the right, so much so that almost all weights in the $\ell_{10}$ solution are non-zero. See Appendix B for more details.

## 4.3 GENERALIZATION ERRORS OF DIFFERENT MIRRORS

We compare the performance of the SMD algorithms discussed before on the test set. For MNIST, perhaps not surprisingly, all the four SMD algorithms achieve around 99% or higher accuracy. For CIFAR-10, however, there is a significant difference between the test errors of different mirrors/regularizers on the same ResNet-18 architecture. Fig. 7 shows the test accuracies of different algorithms with eight random initializations around zero, as discussed before. Counter-intuitively, $\ell_{10}$ performs consistently best, while $\ell_1$ performs consistently worse. This result suggests the importance of a comprehensive study of the role of regularization, and the choice of the best regularizer, to improve the generalization performance of deep neural networks.

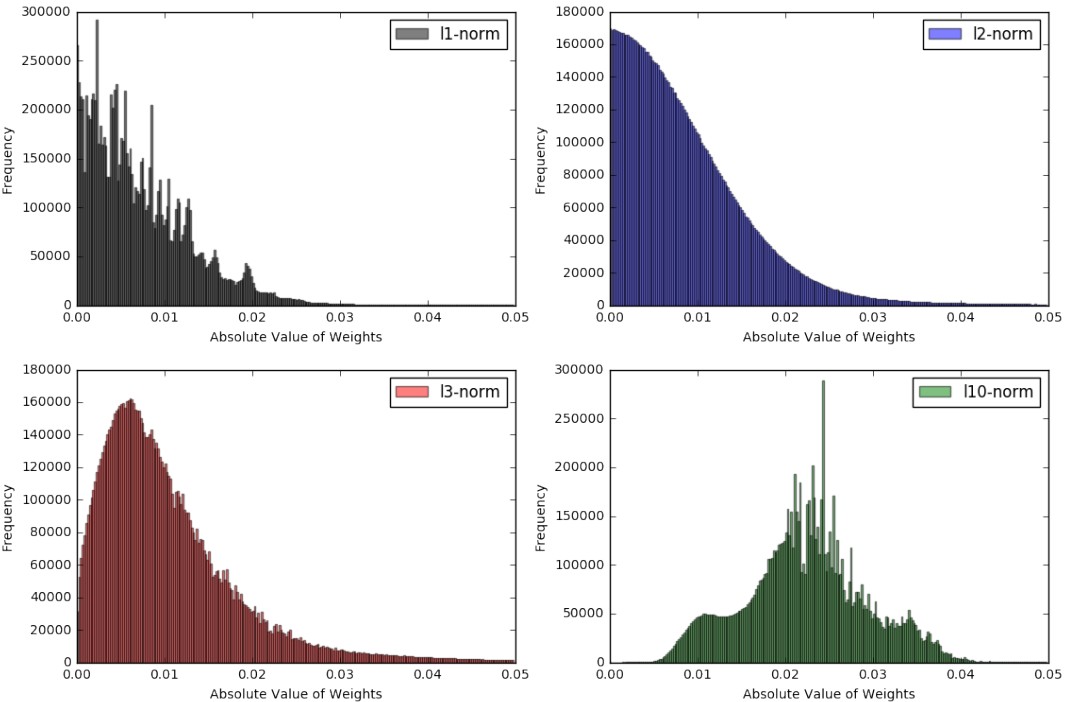

Figure 6: Histogram of the absolute value of the final weights in the network for different SMD algorithm with different potentials. Note that each of the four histograms corresponds to an $11 \times 10^6$-dimensional weight vector that perfectly interpolates the data. Even though the weights remain quite small, the histograms are drastically different. $\ell_1$-SMD induces sparsity on the weights, as expected. SGD appears to be produce a Gaussian distribution on the weights. $\ell_3$-SMD starts to reduce the sparsity, and $\ell_{10}$ shifts the distribution of the weights significantly, so much so that almost all the weights are non-zero.

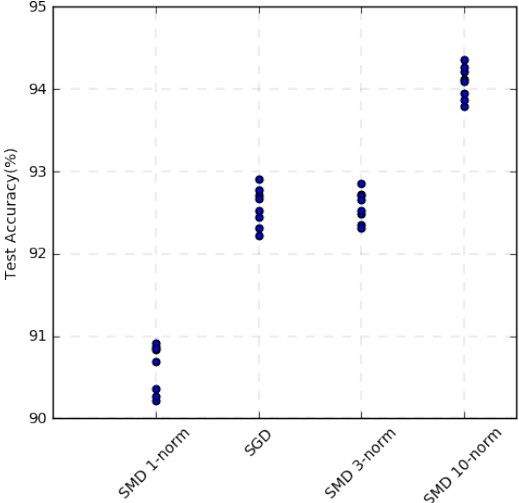

Figure 7: Generalization performance of different SMD algorithms on the CIFAR-10 dataset using ResNet-18. $\ell_{10}$ performs consistently better, while $\ell_1$ performs consistently worse.

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

# Supplementary Material

## A  Proofs of the Theoretical Results

In this section, we prove the main theoretical results. The proofs are based on a fundamental identity about the iterates of SMD, which holds for all mirrors and all overparametereized (even nonlinear) models (Lemma 6). We first prove this identity, and then use it to prove the convergence and implicit regularization results.

### A.1  Fundamental Identity of SMD

Let us prove the fundamental identity.

**Lemma 6.** *For any model $f(\cdot, \cdot)$, any differentiable loss $\ell(\cdot)$, any parameter $w \in \mathcal{W}$, and any step size $\eta > 0$, the following relation holds for the SMD iterates $\{w_i\}$*

$$D_\psi(w, w_{i-1}) = D_\psi(w, w_i) + D_{\psi - \eta L_i}(w_i, w_{i-1}) + \eta L_i(w_i) + \eta D_{L_i}(w, w_{i-1}), \quad (9)$$

*for all $i \geq 1$.*

*Proof of Lemma 6.*  Let us start by expanding the Bregman divergence $D_\psi(w, w_i)$ based on its definition

$$D_\psi(w, w_i) = \psi(w) - \psi(w_i) - \nabla\psi(w_i)^T(w - w_i).$$

By plugging the SMD update rule $\nabla\psi(w_i) = \nabla\psi(w_{i-1}) - \eta\nabla L_i(w_{i-1})$ into this, we can write it as

$$D_\psi(w, w_i) = \psi(w) - \psi(w_i) - \nabla\psi(w_{i-1})^T(w - w_i) + \eta\nabla L_i(w_{i-1})^T(w - w_i). \quad (10)$$

Using the definition of Bregman divergence for $(w, w_{i-1})$ and $(w_i, w_{i-1})$, i.e., $D_\psi(w, w_{i-1}) = \psi(w) - \psi(w_{i-1}) - \nabla\psi(w_{i-1})^T(w - w_{i-1})$ and $D_\psi(w_i, w_{i-1}) = \psi(w_i) - \psi(w_{i-1}) - \nabla\psi(w_{i-1})^T(w_i - w_{i-1})$, we can express this as

$$
\begin{aligned}
D_\psi(w, w_i) = {} & D_\psi(w, w_{i-1}) + \psi(w_{i-1}) + \nabla\psi(w_{i-1})^T(w - w_{i-1}) - \psi(w_i) \\
& \qquad\qquad - \nabla\psi(w_{i-1})^T(w - w_i) + \eta\nabla L_i(w_{i-1})^T(w - w_i) \quad (11) \\
= {} & D_\psi(w, w_{i-1}) + \psi(w_{i-1}) - \psi(w_i) + \nabla\psi(w_{i-1})^T(w_i - w_{i-1}) \\
& \qquad\qquad\qquad\qquad\qquad\qquad\qquad + \eta\nabla L_i(w_{i-1})^T(w - w_i) \quad (12) \\
= {} & D_\psi(w, w_{i-1}) - D_\psi(w_i, w_{i-1}) + \eta\nabla L_i(w_{i-1})^T(w - w_i). \quad (13)
\end{aligned}
$$

Expanding the last term using $w - w_i = (w - w_{i-1}) - (w_i - w_{i-1})$, and following the definition of $D_{L_i}(.,.)$ from (7) for $(w, w_{i-1})$ and $(w_i, w_{i-1})$, we have

$$
\begin{aligned}
D_\psi(w, w_i) = {} & D_\psi(w, w_{i-1}) - D_\psi(w_i, w_{i-1}) + \eta\nabla L_i(w_{i-1})^T(w - w_{i-1}) \\
& \qquad\qquad\qquad\qquad\qquad\qquad\qquad - \eta\nabla L_i(w_{i-1})^T(w_i - w_{i-1}) \quad (14) \\
= {} & D_\psi(w, w_{i-1}) - D_\psi(w_i, w_{i-1}) + \eta\left(L_i(w) - L_i(w_{i-1}) - D_{L_i}(w, w_{i-1})\right) \\
& \qquad\qquad\qquad\qquad - \eta\left(L_i(w_i) - L_i(w_{i-1}) - D_{L_i}(w_i, w_{i-1})\right) \quad (15) \\
= {} & D_\psi(w, w_{i-1}) - D_\psi(w_i, w_{i-1}) + \eta\left(L_i(w) - D_{L_i}(w, w_{i-1})\right) \\
& \qquad\qquad\qquad\qquad\qquad\qquad - \eta\left(L_i(w_i) - D_{L_i}(w_i, w_{i-1})\right) \quad (16)
\end{aligned}
$$

Note that for all $w \in \mathcal{W}$, we have $L_i(w) = 0$. Therefore, for all $w \in \mathcal{W}$

$$D_\psi(w, w_i) = D_\psi(w, w_{i-1}) - D_\psi(w_i, w_{i-1}) - \eta D_{L_i}(w, w_{i-1}) - \eta L_i(w_i) + \eta D_{L_i}(w_i, w_{i-1}). \quad (17)$$

Combining the second and the last terms in the right-hand side leads to

$$D_\psi(w, w_i) = D_\psi(w, w_{i-1}) - D_{\psi - \eta L_i}(w_i, w_{i-1}) - \eta D_{L_i}(w, w_{i-1}) - \eta L_i(w_i), \quad (18)$$

for all $w \in \mathcal{W}$, which concludes the proof.  $\square$

A.2   CONVERGENCE OF SMD TO THE INTERPOLATING SET

Now that we have proved Lemma 6, we can use it to prove our main results, in a remarkably simple fashion. Let us first prove the convergence of SMD to the set of solutions.

**Assumption 1.** *Denote the initial point by $w_0$. There exists $w \in \mathcal{W}$ and a region $\mathcal{B} = \{w' \in \mathbb{R}^p \mid D_\psi(w, w') \leq \epsilon\}$ containing $w_0$, such that $D_{L_i}(w, w') \geq 0, i = 1, \ldots, n$, for all $w' \in \mathcal{B}$.*

**Theorem 3.** *Consider the set of interpolating parameters $\mathcal{W} = \{w \in \mathbb{R}^p \mid f(x_i, w) = y_i, i = 1, \ldots, n\}$, and the SMD iterates given in (3), where every data point is revisited after some steps. Under Assumption 1, for sufficiently small step size, i.e., for any $\eta > 0$ for which $\psi(\cdot) - \eta L_i(\cdot)$ is strictly convex for all $i$, the following holds.*

1. *All the iterates $\{w_i\}$ remain in $\mathcal{B}$.*

2. *The iterates converge (to $w_\infty$).*

3. *$w_\infty \in \mathcal{W}$.*

*Proof of Theorem 3.* First we show that all the iterates wil remain in $\mathcal{B}$. Recall the identity of SMD from Lemma 6:

$$D_\psi(w, w_{i-1}) = D_\psi(w, w_i) + D_{\psi - \eta L_i}(w_i, w_{i-1}) + \eta L_i(w_i) + \eta D_{L_i}(w, w_{i-1}) \tag{9}$$

which holds for all $w \in \mathcal{W}$. If $w_{i-1}$ is in the region $\mathcal{B}$, we know that the last term $D_{L_i}(w, w_{i-1})$ is non-negative. Furthermore, if the step size is small enough that $\psi(\cdot) - \eta L_i(\cdot)$ is strictly convex, the second term $D_{\psi - \eta L_i}(w_i, w_{i-1})$ is a Bregman divergence and is non-negative. Since the loss is non-negative, $\eta L_i(w_i)$ is always non-negative. As a result, we have

$$D_\psi(w, w_{i-1}) \geq D_\psi(w, w_i), \tag{19}$$

This implies that $D_\psi(w, w_i) \leq \epsilon$, which means $w_i$ is in $\mathcal{B}$ too. Since $w_0$ is in $\mathcal{B}$, $w_1$ will be in $\mathcal{B}$, and therefore, $w_2$ will be in $\mathcal{B}$, and similarly all the iterates will remain in $\mathcal{B}$.

Next, we prove that the iterates converge and $w_\infty \in \mathcal{W}$. If we sum up identity (9) for all $i = 1, \ldots, T$, the first terms on the right- and left-hand side cancel each other telescopically, and we have

$$D_\psi(w, w_0) = D_\psi(w, w_T) + \sum_{i=1}^{T} \left[ D_{\psi - \eta L_i}(w_i, w_{i-1}) + \eta L_i(w_i) + \eta D_{L_i}(w, w_{i-1}) \right]. \tag{20}$$

Since $D_\psi(w, w_T) \geq 0$, we have $\sum_{i=1}^{T} \left[ D_{\psi - \eta L_i}(w_i, w_{i-1}) + \eta L_i(w_i) + \eta D_{L_i}(w, w_{i-1}) \right] \leq D_\psi(w, w_0)$. If we take $T \to \infty$, the sum still has to remain bounded, i.e.,

$$\sum_{i=1}^{\infty} \left[ D_{\psi - \eta L_i}(w_i, w_{i-1}) + \eta L_i(w_i) + \eta D_{L_i}(w, w_{i-1}) \right] \leq D_\psi(w, w_0). \tag{21}$$

Since the step size is small enough that $\psi(\cdot) - \eta L_i(\cdot)$ is strictly convex for all $i$, the first term $D_{\psi - \eta L_i}(w_i, w_{i-1})$ is non-negative. The second term $\eta L_i(w_i)$ is non-negative because of the non-negativity of the loss. Finally, the last term $D_{L_i}(w, w_{i-1})$ is non-negative because $w_{i-1} \in \mathcal{B}$ for all $i$. Hence, all the three terms in the summand are non-negative, and because the sum is bounded, they should go to zero as $i \to \infty$. In particular,

$$D_{\psi - \eta L_i}(w_i, w_{i-1}) \to 0 \tag{22}$$

implies $w_i \to w_{i-1}$, i.e., convergence ($w_i \to w_\infty$) (Note that the functions $\psi - \eta L_i$ do not go to zero, as there is a fixed number, i.e., $n$, of them). Further,

$$\eta L_i(w_i) \to 0. \tag{23}$$

This implies that all the individual losses are going to zero, and since every data point is being revisited after some steps, all the data points are being fit. Therefore, $w_\infty \in \mathcal{W}$.  □

A.3   CLOSENESS OF THE FINAL POINT TO THE REGULARIZED SOLUTION

In this section, we show that with the additional Assumption 2 (which is equivalent to $f_i(\cdot)$ having bounded Hessian in $\mathcal{B}$), not only do the iterates remain in $\mathcal{B}$ and converge to the set $\mathcal{W}$, but also they converge to a point which is very close to $w^*$ (the closest solution to the initial point, in Bregman divergence). The proof is again based on our fundamental identity for SMD.

**Assumption 2.** *Consider the region $\mathcal{B}$ in Assumption 1. $f_i(\cdot)$ have bounded gradient and Hessian on the convex hull of $\mathcal{B}$, i.e., $\|\nabla f_i(w')\| \le \gamma$, and $\alpha \le \lambda_{\min}(H_{f_i}(w')) \le \lambda_{\max}(H_{f_i}(w')) \le \beta, i = 1, \dots, n$, for all $w' \in \operatorname{conv} \mathcal{B}$.*

**Theorem 4.** *Define $w^* = \arg\min_{w \in \mathcal{W}} D_\psi(w, w_0)$. Under the assumptions of Theorem 3, and Assumption 2, the following holds.*

   1. $D_\psi(w_\infty, w_0) = D_\psi(w^*, w_0) + o(\epsilon)$

   2. $D_\psi(w^*, w_\infty) = o(\epsilon)$

*Proof of Theorem 4.* Recall the identity of SMD from Lemma 6:

$$D_\psi(w, w_{i-1}) = D_\psi(w, w_i) + D_{\psi - \eta L_i}(w_i, w_{i-1}) + \eta L_i(w_i) + \eta D_{L_i}(w, w_{i-1}) \qquad (9)$$

which holds for all $w \in \mathcal{W}$. Summing the identity for all $i \ge 1$, we have

$$D_\psi(w, w_0) = D_\psi(w, w_\infty) + \sum_{i=1}^{\infty} \left[ D_{\psi - \eta L_i}(w_i, w_{i-1}) + \eta L_i(w_i) + \eta D_{L_i}(w, w_{i-1}) \right]. \qquad (24)$$

for all $w \in \mathcal{W}$. Note that the only terms in the right-hand side which depend on $w$ are the first one $D_\psi(w, w_\infty)$ and the last one $\eta \sum_{i=1}^{\infty} D_{L_i}(w, w_{i-1})$. In what follows, We will argue that, within $\mathcal{B}$, the dependence on $w$ in the last term is weak and therefore $w_\infty$ is close to $w^*$.

To further spell out the dependence on $w$ in the last term, let us expand $D_{L_i}(w, w_{i-1})$

$$D_{L_i}(w, w_{i-1}) = 0 - L_i(w_{i-1}) - \nabla L_i(w_{i-1})^T (w - w_{i-1}) \qquad (25)$$

$$= -L_i(w_{i-1}) + \ell'(y_i - f_i(w_{i-1}))\nabla f_i(w_{i-1}))^T (w - w_{i-1}) \qquad (26)$$

for all $w \in \mathcal{W}$, where the first equality comes from the definition of $D_{L_i}(\cdot, \cdot)$ and the fact that $L_i(w) = 0$ for $w \in \mathcal{W}$. The second equality is from taking the derivative of $L_i(\cdot) = \ell(y_i - f_i(\cdot))$ and evaluating it at $w_{i-1}$.

By Taylor expansion of $f_i(w)$ around $w_{i-1}$ and using Taylor's theorem (Lagrange's mean-value form), we have

$$f_i(w) = f_i(w_{i-1}) + \nabla f_i(w_{i-1})^T (w - w_{i-1}) + \frac{1}{2}(w - w_{i-1})^T H_{f_i}(\hat{w}_i)(w - w_{i-1}), \qquad (27)$$

for some $\hat{w}_i$ in the convex hull of $w$ and $w_{i-1}$. Since $f_i(w) = y_i$ for all $w \in \mathcal{W}$, it follows that

$$\nabla f_i(w_{i-1})^T (w - w_{i-1}) = y_i - f_i(w_{i-1}) - \frac{1}{2}(w - w_{i-1})^T H_{f_i}(\hat{w}_i)(w - w_{i-1}), \qquad (28)$$

for all $w \in \mathcal{W}$. Plugging this into (26), we have

$$D_{L_i}(w, w_{i-1}) = -L_i(w_{i-1}) + \ell'(y_i - f_i(w_{i-1}))\left(y_i - f_i(w_{i-1}) - \frac{1}{2}(w - w_{i-1})^T H_{f_i}(\hat{w}_i)(w - w_{i-1})\right) \qquad (29)$$

for all $w \in \mathcal{W}$. Finally, by plugging this back into the identity (24), we have

$$D_\psi(w, w_0) = D_\psi(w, w_\infty) + \sum_{i=1}^{\infty} \Big[ D_{\psi - \eta L_i}(w_i, w_{i-1}) + \eta L_i(w_i) - \eta L_i(w_{i-1})$$

$$+ \eta \ell'(y_i - f_i(w_{i-1}))\big(y_i - f_i(w_{i-1}) - \frac{1}{2}(w - w_{i-1})^T H_{f_i}(\hat{w}_i)(w - w_{i-1})\big)\Big]. \qquad (30)$$

for all $w \in \mathcal{W}$. Note that this can be expressed as

$$D_\psi(w, w_0) = D_\psi(w, w_\infty) + C - \sum_{i=1}^{\infty} \frac{1}{2}\eta \ell'(y_i - f_i(w_{i-1}))(w - w_{i-1})^T H_{f_i}(\hat{w}_i)(w - w_{i-1}), \qquad (31)$$

for all $w \in \mathcal{W}$, where $C$ does not depend on $w$:

$$C = \sum_{i=1}^{\infty} \left[ D_{\psi - \eta L_i}(w_i, w_{i-1}) + \eta L_i(w_i) - \eta L_i(w_{i-1}) + \eta \ell'(y_i - f_i(w_{i-1}))(y_i - f_i(w_{i-1})) \right].$$

From Theorem 3, we know that $w_\infty \in \mathcal{W}$. Therefore, by plugging it into equation (31), and using the fact that $D_\psi(w_\infty, w_\infty) = 0$, we have

$$D_\psi(w_\infty, w_0) = C - \sum_{i=1}^{\infty} \frac{1}{2} \eta \ell'(y_i - f_i(w_{i-1}))(w_\infty - w_{i-1})^T H_{f_i}(w_i')(w_\infty - w_{i-1}), \qquad (32)$$

where $w_i'$ is a point in the convex hull of $w_\infty$ and $w_{i-1}$ (and therefore also in conv $\mathcal{B}$), for all $i$. Similarly, by plugging $w^*$, which is also in $\mathcal{W}$, into (31), we have

$$D_\psi(w^*, w_0) = D_\psi(w^*, w_\infty) + C - \sum_{i=1}^{\infty} \frac{1}{2} \eta \ell'(y_i - f_i(w_{i-1}))(w^* - w_{i-1})^T H_{f_i}(w_i'')(w^* - w_{i-1}),$$
$$(33)$$

where $w_i''$ is a point in the convex hull of $w^*$ and $w_{i-1}$ (and therefore also in conv $\mathcal{B}$), for all $i$. Subtracting the last two equations from each other yields

$$D_\psi(w_\infty, w_0) - D_\psi(w^*, w_0) = -D_\psi(w^*, w_\infty) + \sum_{i=1}^{\infty} \frac{1}{2} \eta \ell'(y_i - f_i(w_{i-1})) \cdot$$
$$\left[ (w^* - w_{i-1})^T H_{f_i}(w_i'')(w^* - w_{i-1}) - (w_\infty - w_{i-1})^T H_{f_i}(w_i')(w_\infty - w_{i-1}) \right]. \quad (34)$$

Note that since all $w_i'$ and $w_i''$ are in conv $\mathcal{B}$, by Assumption 2, we have

$$\alpha \|w_\infty - w_{i-1}\|^2 \leq (w_\infty - w_{i-1})^T H_{f_i}(w_i')(w_\infty - w_{i-1}) \leq \beta \|w_\infty - w_{i-1}\|^2, \qquad (35)$$

and

$$\alpha \|w^* - w_{i-1}\|^2 \leq (w^* - w_{i-1})^T H_{f_i}(w_i'')(w^* - w_{i-1}) \leq \beta \|w^* - w_{i-1}\|^2. \qquad (36)$$

Further, again since all the iterates $\{w_i\}$ are in $\mathcal{B}$, it follows that $\|w_\infty - w_{i-1}\|^2 = O(\epsilon)$ and $\|w^* - w_{i-1}\|^2 = O(\epsilon)$. As a result the difference of the two terms, i.e., $\left[ (w^* - w_{i-1})^T H_{f_i}(w_i'')(w^* - w_{i-1}) - (w_\infty - w_{i-1})^T H_{f_i}(w_i')(w_\infty - w_{i-1}) \right]$, is also $O(\epsilon)$, and we have

$$D_\psi(w_\infty, w_0) - D_\psi(w^*, w_0) = -D_\psi(w^*, w_\infty) + \sum_{i=1}^{\infty} \eta \ell'(y_i - f_i(w_{i-1}))O(\epsilon). \qquad (37)$$

Now note that $\ell'(y_i - f_i(w_{i-1})) = \ell'(f_i(w) - f_i(w_{i-1})) = \ell'(\nabla f_i(\tilde{w}_i)^T(w - w_{i-1}))$ for some $\tilde{w}_i \in$ conv $\mathcal{B}$. Since $\|w - w_{i-1}\|^2 = O(\epsilon)$ for all $i$, and since $\ell(\cdot)$ is differentiable and $f_i(\cdot)$ have bounded derivatives, it follows that $\ell'(y_i - f_i(w_{i-1})) = o(\epsilon)$. Furthermore, the sum is bounded. This implies that $D_\psi(w_\infty, w_0) - D_\psi(w^*, w_0) = -D_\psi(w^*, w_\infty) + o(\epsilon)$, or equivalently

$$\left( D_\psi(w_\infty, w_0) - D_\psi(w^*, w_0) \right) + D_\psi(w^*, w_\infty) = o(\epsilon). \qquad (38)$$

The term in parentheses $D_\psi(w_\infty, w_0) - D_\psi(w^*, w_0)$ is non-negative by definition of $w^*$. The second term $D_\psi(w^*, w_\infty)$ is non-negative by convexity of $\psi$. Since both terms are non-negative and their sum is $o(\epsilon)$, each one of them is at most $o(\epsilon)$, i.e.

$$\begin{cases} D_\psi(w_\infty, w_0) - D_\psi(w^*, w_0) = o(\epsilon) \\ D_\psi(w^*, w_\infty) = o(\epsilon) \end{cases} \qquad (39)$$

which concludes the proof. $\qquad \square$

**Corollary 5.** *For the initialization* $w_0 = \arg\min_{w \in \mathbb{R}^p} \psi(w)$, *under the conditions of Theorem 4,* $w^* = \arg\min_{w \in \mathcal{W}} \psi(w)$ *and the following holds.*

1. $\psi(w_\infty) = \psi(w^*) + o(\epsilon)$

2. $D_\psi(w^*, w_\infty) = o(\epsilon)$

*Proof of Corollary 5.* The proof is a straightforward application of Theorem 4. Note that we have

$$D_\psi(w, w_0) = \psi(w) - \psi(w_0) - \nabla\psi(w_0)^T(w - w_0) \tag{40}$$

for all $w$. When $w_0 = \arg\min_{w \in \mathbb{R}^p} \psi(w)$, it follows that $\nabla\psi(w_0) = 0$, and

$$D_\psi(w, w_0) = \psi(w) - \psi(w_0). \tag{41}$$

In particular, by plugging in $w_\infty$ and $w^*$, we have $D_\psi(w_\infty, w_0) = \psi(w_\infty) - \psi(w_0)$ and $D_\psi(w^*, w_0) = \psi(w^*) - \psi(w_0)$. Subtracting the two equations from each other yields

$$D_\psi(w_\infty, w_0) - D_\psi(w^*, w_0) = \psi(w_\infty) - \psi(w^*), \tag{42}$$

which along with the application of Theorem 4 concludes the proof. □

### A.4 Closeness to the Interpolating Set in Highly Overparameterized Models

As we mentioned earlier, it has been argued in a number of recent papers that for highly overparameterized models, any random initial point is, whp, close to the solution set $\mathcal{W}$ (Azizan & Hassibi, 2019; Li & Liang, 2018; Du et al., 2018; Allen-Zhu et al., 2019; Cao & Gu, 2019). In the highly overparameterized regime, $p \gg n$, and so the dimension of the manifold $\mathcal{W}$, which is $p - n$, is very large. For simplicity, we outline an argument for the case of Euclidean distance, bearing in mind that a similar argument can be used for general Bregman divergence. Note that the distance of an arbitrarily chosen $w_0$ to $\mathcal{W}$ is given by

$$\min_w \quad \|w - w_0\|^2$$
$$\text{s.t.} \quad y = f(x, w)$$

where $y = \text{vec}(y_i, i = 1, \ldots, n)$ and $f(x, w) = \text{vec}(f(x_i, w), i = 1, \ldots, n)$. This can be approximated by

$$\min_w \quad \|w - w_0\|^2$$
$$\text{s.t.} \quad y \approx f(x, w_0) + \nabla f(x, w_0)^T(w - w_0)$$

where $\nabla f(x, w_0)^T = \text{vec}(\nabla f(x_i, w)^T, i = 1, \ldots, n)$ is the $n \times p$ Jacobian matrix. The latter optimization can be solved to yield

$$\|w_* - w_0\|^2 \approx (y - f(x, w_0))^T \left(\nabla f(x, w_0)^T \nabla f(x, w_0)\right)^{-1} (y - f(x, w_0)) \tag{43}$$

Note that $\nabla f(x, w_0)^T \nabla f(x, w_0)$ is an $n \times n$ matrix consisting of the sum of $p$ outer products. When the $x_i$ are sufficiently random, and $p \gg n$, it is not unreasonable to assume that whp

$$\lambda_{\min}\left(\nabla f(x, w_0)^T \nabla f(x, w_0)\right) = \Omega(p),$$

from which we conclude

$$\|w_* - w_0\|^2 \approx \|y - f(x, w_0)\|^2 \cdot O(\frac{1}{p}) = O(\frac{n}{p}), \tag{44}$$

since $y - f(x, w_0)$ is $n$-dimensional. The above implies that $w_0$ is close to $w_*$ and hence $\mathcal{W}$.

## B    MORE DETAILS ON THE EXPERIMENTAL RESULTS

In order to evaluate the claim, we run systematic experiments on some standard deep learning problems.

**Datasets.** We use the standard MNIST (LeCun et al., 1998) and CIFAR-10 (Krizhevsky & Hinton, 2009) datasets.

**Architectures.** For MNIST, we use a 4-layer convolutional neural network (CNN) with 2 convolution layers and 2 fully connected layers. The convolutional layers and the fully connected layers are picked wide enough to obtain $2 \times 10^6$ trainable parameters. Since MNIST dataset has 60,000 training samples, the number of parameters is significantly larger than the number of training data points, and the problem is highly overparameterized. For the CIFAR-10 dataset, we use the standard ResNet-18 (He et al., 2016) architecture without any modifications. CIFAR-10 has 50,000 training samples and with the total number of $11 \times 10^6$ parameters in ResNet-18, the problem is again highly overparameterized.

**Loss Function.** We use the cross-entropy loss as the loss function in our training. We train the models from different initializations, and with different mirror descents from each particular initialization, until we reach $100\%$ training accuracy, i.e., until we hit $\mathcal{W}$.

**Initialization.** We randomly initialize the parameters of the networks around zero ($\mathcal{N}(0, 0.0001)$). We choose 6 independent initializations for the CNN, and 8 for ResNet-18, and for each initialization, we run the following 4 different SMD algorithms.

**Algorithms.** We use the mirror descent algorithms defined by the norm potential $\psi(w) = \frac{1}{q}\|w\|_q^q$ for the following four different norms: (a) $\ell_1$ norm, i.e., $q = 1 + \epsilon$, (b) $\ell_2$ norm, i.e., $q = 2$ (which is SGD), (c) $\ell_3$ norm, i.e., $q = 3$, (d) $\ell_{10}$ norm, i.e., $q = 10$ (as a surrogate for $\ell_\infty$ norm). The update rule can be expressed as follows.

$$w_{i,j} = \left| |w_{i-1,j}|^{q-1} \operatorname{sign}(w_{i-1,j}) - \eta \nabla L_i(w_{i-1})_j \right|^{\frac{1}{q-1}} \cdot$$

$$\operatorname{sign}\left( |w_{i-1,j}|^{q-1} \operatorname{sign}(w_{i-1,j}) - \eta \nabla L_i(w_{i-1})_j \right), \quad (45)$$

where $w_{i-1,j}$ denotes the $j$-th element of the $w_{i-1}$ vector.

We use a fixed step size $\eta$. The step size is chosen to obtain convergence to global minima.

### B.1    MNIST EXPERIMENTS

#### B.1.1    CLOSEST MINIMUM FOR DIFFERENT MIRROR DESCENTS WITH FIXED INITIALIZATION

We provide the distances from final points (global minima) obtained by different algorithms from the same initialization, measured in different Bregman divergences for MNIST classification task using a standard CNN. Note that in all tables the smallest element in each row is on the diagonal, which means the point achieved by each mirror has the smallest Bregman divergence to the initialization corresponding to that mirror, among all mirrors. Tables 3, 4, 5, 6, 7, 8 depict these results for 6 different initializations. The rows are the distance metrics used as the Bregman Divergences with specified potentials. The columns are the global minima obtained using specified SMD algorithms.

Table 3: MNIST Initial Point 1

|  | SMD 1-norm | SMD 2-norm (SGD) | SMD 3-norm | SMD 10-norm |
|---|---|---|---|---|
| 1-norm BD | 2.767 | 937.8 | $1.05 \times 10^4$ | $1.882 \times 10^5$ |
| 2-norm BD | 301.6 | 58.61 | 261.3 | $2.118 \times 10^4$ |
| 3-norm BD | 1720 | 37.45 | 7.143 | 2518 |
| 10-norm BD | $7.453 \times 10^8$ | 773.4 | 0.2939 | 0.003545 |

Table 4: MNIST Initial Point 2

|            | SMD 1-norm           | SMD 2-norm (SGD) | SMD 3-norm         | SMD 10-norm          |
|------------|----------------------|------------------|--------------------|----------------------|
| 1-norm BD  | 2.78                 | 945              | $1.37 \times 10^4$ | $2.01 \times 10^5$   |
| 2-norm BD  | 292                  | 59.3             | 374                | $2.29 \times 10^4$   |
| 3-norm BD  | $1.51 \times 10^3$   | 38.6             | 11.6               | $2.71 \times 10^3$   |
| 10-norm BD | $1.06 \times 10^8$   | 831              | 0.86               | 0.00321              |

Table 5: MNIST Initial Point 3

|            | SMD 1-norm           | SMD 2-norm (SGD) | SMD 3-norm         | SMD 10-norm          |
|------------|----------------------|------------------|--------------------|----------------------|
| 1-norm BD  | 3.02                 | 968              | $1.06 \times 10^4$ | $1.9 \times 10^5$    |
| 2-norm BD  | 291                  | 60.9             | 272                | $2.12 \times 10^4$   |
| 3-norm BD  | $1.49 \times 10^3$   | 39.1             | 7.82               | $2.49 \times 10^3$   |
| 10-norm BD | $1.1 \times 10^8$    | 900              | 0.411              | 0.00318              |

Table 6: MNIST Initial Point 4

|            | SMD 1-norm           | SMD 2-norm (SGD)   | SMD 3-norm         | SMD 10-norm          |
|------------|----------------------|--------------------|--------------------|----------------------|
| 1-norm BD  | 2.78                 | $1.21 \times 10^3$ | $1.08 \times 10^4$ | $1.92 \times 10^5$   |
| 2-norm BD  | 291                  | 77.3               | 271                | $2.15 \times 10^4$   |
| 3-norm BD  | $1.48 \times 10^3$   | 49.7               | 7.56               | $2.52 \times 10^3$   |
| 10-norm BD | $9.9 \times 10^7$    | $1.72 \times 10^3$ | 0.352              | 0.00296              |

Table 7: MNIST Initial Point 5

|            | SMD 1-norm           | SMD 2-norm (SGD) | SMD 3-norm         | SMD 10-norm          |
|------------|----------------------|------------------|--------------------|----------------------|
| 1-norm BD  | 2.79                 | 958              | $1.08 \times 10^4$ | $2 \times 10^5$      |
| 2-norm BD  | 292                  | 60.4             | 271                | $2.28 \times 10^4$   |
| 3-norm BD  | $1.49 \times 10^3$   | 39               | 7.52               | $2.69 \times 10^3$   |
| 10-norm BD | $9.09 \times 10^7$   | 846              | 0.342              | 0.00309              |

Table 8: MNIST Initial Point 6

|            | SMD 1-norm           | SMD 2-norm (SGD) | SMD 3-norm         | SMD 10-norm          |
|------------|----------------------|------------------|--------------------|----------------------|
| 1-norm BD  | 2.96                 | 930              | $1.08 \times 10^4$ | $1.9 \times 10^5$    |
| 2-norm BD  | 308                  | 59               | 271                | $2.12 \times 10^4$   |
| 3-norm BD  | $1.63 \times 10^3$   | 38.6             | 7.46               | $2.47 \times 10^3$   |
| 10-norm BD | $1.65 \times 10^8$   | 864              | 0.334              | 0.00295              |

### B.1.2 CLOSEST MINIMUM FOR DIFFERENT INITILIZATIONS WITH FIXED MIRROR

We provide the pairwise distances between different initial points and the final points (global minima) obtained by using fixed SMD algorithms in MNIST dataset using a standard CNN. Note that the smallest element in each row is on the diagonal, which means the closest final point to each initialization, among all the final points, is the one corresponding to that point. Tables 9, 10, 11 and 12 depict these results for 4 different SMD algorithms. The rows are the initial points and the columns are the final points corresponding to each initialization.

### B.1.3 CLOSEST MINIMUM FOR DIFFERENT INITILIZATIONS AND DIFFERENT MIRRORS

Now we assess the pairwise distances between different initial points and final points (global minima) obtained by all different initilizations and all different mirrors (Table 8). The smallest element in each row is exactly the final point obtained by that mirror from that initialization, among all the mirrors and all the initial points.

Table 9: MNIST 1-norm Bregman Divergence Between the Initial Points and the Final Points obtained by SMD 1-norm

|                 | Final 1 | Final 2 | Final 3 | Final 4 | Final 5 | Final 6 |
|-----------------|---------|---------|---------|---------|---------|---------|
| Initial Point 1 | 2.7671  | 20311   | 20266   | 20331   | 20340   | 20282   |
| Initial Point 2 | 20332   | 2.7774  | 20281   | 20299   | 20312   | 20323   |
| Initial Point 3 | 20319   | 20312   | 3.018   | 20344   | 20309   | 20322   |
| Initial Point 4 | 20339   | 20279   | 20310   | 2.781   | 20321   | 20297   |
| Initial Point 5 | 20347   | 20317   | 20273   | 20316   | 2.7902  | 20311   |
| Initial Point 6 | 20344   | 20323   | 20340   | 20318   | 20321   | 2.964   |

Table 10: MNIST 2-norm Bregman Divergence Between the Initial Points and the Final Points obtained by SMD 2-norm (SGD)

|                 | Final 1 | Final 2 | Final 3 | Final 4 | Final 5 | Final 6 |
|-----------------|---------|---------|---------|---------|---------|---------|
| Initial Point 1 | 58.608  | 670.75  | 667.03  | 684.18  | 671.36  | 667.84  |
| Initial Point 2 | 669.84  | 59.315  | 669.16  | 682.04  | 669.45  | 669.98  |
| Initial Point 3 | 666.35  | 670.22  | 60.858  | 683.44  | 667.57  | 669.99  |
| Initial Point 4 | 669.71  | 668.86  | 671.19  | 77.275  | 670.33  | 669.7   |
| Initial Point 5 | 671.1   | 669.12  | 668.45  | 683.61  | 60.39   | 666.04  |
| Initial Point 6 | 669.46  | 670.92  | 671.59  | 684.32  | 667.37  | 59.043  |

Table 11: MNIST 3-norm Bregman Divergence Between the Initial Points and the Final Points obtained by SMD 3-norm

|                 | Final 1 | Final 2 | Final 3 | Final 4 | Final 5 | Final 6 |
|-----------------|---------|---------|---------|---------|---------|---------|
| Initial Point 1 | 7.143   | 35.302  | 32.077  | 32.659  | 32.648  | 32.309  |
| Initial Point 2 | 32.507  | 11.578  | 32.256  | 32.325  | 32.225  | 32.46   |
| Initial Point 3 | 31.594  | 34.643  | 7.8239  | 32.521  | 31.58   | 32.519  |
| Initial Point 4 | 32.303  | 34.811  | 32.937  | 7.5589  | 32.617  | 32.284  |
| Initial Point 5 | 32.673  | 34.678  | 32.071  | 32.738  | 7.5188  | 31.558  |
| Initial Point 6 | 32.116  | 34.731  | 32.376  | 32.431  | 31.699  | 7.4593  |

Table 12: MNIST 10-norm Bregman Divergence Between the Initial Points and the Final Points obtained by SMD 10-norm

|                 | Final 1 | Final 2 | Final 3 | Final 4 | Final 5 | Final 6 |
|-----------------|---------|---------|---------|---------|---------|---------|
| Initial Point 1 | 0.00354 | 0.37    | 0.403   | 0.286   | 0.421   | 0.408   |
| Initial Point 2 | 0.33    | 0.00321 | 0.369   | 0.383   | 0.415   | 0.422   |
| Initial Point 3 | 0.347   | 0.318   | 0.00318 | 0.401   | 0.312   | 0.406   |
| Initial Point 4 | 0.282   | 0.38    | 0.458   | 0.00296 | 0.491   | 0.376   |
| Initial Point 5 | 0.405   | 0.418   | 0.354   | 0.484   | 0.00309 | 0.48    |
| Initial Point 6 | 0.403   | 0.353   | 0.422   | 0.331   | 0.503   | 0.00295 |

| | F1 SMD 1 | F2 SMD 1 | F3 SMD 1 | F4 SMD 1 | F5 SMD 1 | F6 SMD 1 | F1 SMD 2 | F2 SMD 2 | F3 SMD 2 | F4 SMD 2 | F5 SMD 2 | F6 SMD 2 | F1 SMD 3 | F2 SMD 3 | F3 SMD 3 | F4 SMD 3 | F5 SMD 3 | F6 SMD 3 | F1 SMD 10 | F2 SMD 10 | F3 SMD 10 | F4 SMD 10 | F5 SMD 10 | F6 SMD 10 |
|---|---|---|---|---|---|---|---|---|---|---|---|---|---|---|---|---|---|---|---|---|---|---|---|---|
| l1 1-norm BD | 2.767105 | 20310.58 | 20266.27 | 20330.6 | 20340.2 | 20281.51 | 937.7902 | 20501.09 | 20453.6 | 20615.37 | 20505.63 | 20451.42 | 10500.44 | 24298.6 | 22690.41 | 22883.13 | 22928.17 | 22930.01 | 188233.4 | 200749.8 | 189599.3 | 192017.6 | 200332.6 | 189842.1 |
| l2 1-norm BD | 20332.47 | 2.777443 | 20280.59 | 20298.8 | 20312 | 20322.66 | 20477.15 | 944.8926 | 20467.58 | 20572.54 | 20486.79 | 20481.46 | 22902.71 | 13736.89 | 22683.03 | 22823.09 | 22927.75 | 22951.2 | 188019 | 200838.7 | 189406.7 | 191694.7 | 200319.4 | 189452.9 |
| l3 1-norm BD | 20319.38 | 20312.19 | 3.018036 | 20343.8 | 20308.9 | 20322.02 | 20443.74 | 20487.21 | 967.6324 | 20612.98 | 20486.93 | 20485.8 | 22897.06 | 24300.62 | 10609.4 | 22876.31 | 22901.84 | 22949.55 | 187883.2 | 201071.8 | 189571 | 192131.8 | 199958.1 | 189571.5 |
| l4 1-norm BD | 20338.77 | 20279.16 | 20309.78 | 2.78104 | 20321.1 | 20297.36 | 20476.14 | 20461.38 | 20499.16 | 1214.917 | 20499.88 | 20469.45 | 22910.51 | 24283.22 | 22733.45 | 10756.58 | 22928.43 | 22938.72 | 187740.6 | 200692.5 | 189522.4 | 192082.9 | 200434.4 | 189653.4 |
| l5 1-norm BD | 20347.03 | 20317.23 | 20273.07 | 20316.4 | 2.79019 | 20310.78 | 20498.73 | 20496.97 | 20464.54 | 20600.07 | 957.8013 | 20484.67 | 22921.69 | 24335.41 | 22722.83 | 22877.07 | 10812.1 | 22955.94 | 188056.5 | 200743.9 | 189707.6 | 192056.4 | 200478.6 | 189883 |
| l6 1-norm BD | 20343.59 | 20322.62 | 20339.82 | 20318.4 | 20320.9 | 2.964027 | 20493.68 | 20504.14 | 20535.06 | 20590.71 | 20491.61 | 930.2714 | 22926.8 | 24311.73 | 22713.74 | 22837.8 | 22900.31 | 10848.27 | 187959.4 | 200482.2 | 189602.3 | 192052.5 | 200309.6 | 189738.7 |
| l1 2-norm BD | 301.6218 | 928.1953 | 922.246 | 925.889 | 929.909 | 940.8018 | 58.60796 | 670.7482 | 667.0325 | 684.1751 | 671.3561 | 667.8379 | 261.2823 | 760.2361 | 701.1998 | 706.1766 | 704.5516 | 704.0641 | 21179.18 | 22902.48 | 21188.34 | 21536.64 | 22803.44 | 21162.22 |
| l2 2-norm BD | 938.5225 | 291.6223 | 924.8324 | 925.561 | 926.34 | 944.1569 | 669.8414 | 59.31496 | 669.1617 | 682.0358 | 669.4517 | 669.9774 | 703.9956 | 373.9718 | 702.4789 | 703.8165 | 703.9578 | 705.4268 | 21164.67 | 22901.68 | 21187.33 | 21523.37 | 22797.37 | 21152.21 |
| l3 2-norm BD | 936.4615 | 928.4752 | 290.902 | 926.259 | 924.438 | 943.7131 | 666.3494 | 670.2202 | 60.85767 | 683.4393 | 667.5668 | 669.9933 | 700.8777 | 758.6538 | 272.0649 | 705.6848 | 701.1583 | 705.155 | 21164.46 | 22904.93 | 21186.56 | 21536.03 | 22787.11 | 21151.72 |
| l4 2-norm BD | 938.7566 | 926.655 | 926.2601 | 290.552 | 928.945 | 944.0035 | 669.7086 | 668.8569 | 671.186 | 77.27538 | 670.3311 | 669.7023 | 703.3976 | 757.9133 | 704.6345 | 270.9099 | 703.842 | 704.6346 | 21161.99 | 22898.71 | 21186.54 | 21541.32 | 22799.92 | 21152.3 |
| l5 2-norm BD | 940.8469 | 928.4445 | 923.667 | 927.336 | 291.765 | 939.7045 | 671.1005 | 669.1169 | 668.446 | 683.6102 | 60.39 | 666.0443 | 705.3977 | 758.6884 | 702.3112 | 705.3715 | 270.8719 | 701.3619 | 21166.8 | 22898.1 | 21191.65 | 21533.87 | 22805.13 | 21162.55 |
| l6 2-norm BD | 937.93 | 929.7885 | 929.404 | 927.348 | 925.181 | 307.5172 | 669.4556 | 670.9225 | 671.5908 | 684.3248 | 667.3748 | 59.04266 | 702.8038 | 759.7584 | 703.6673 | 705.2996 | 700.8271 | 271.1133 | 21166.69 | 22894.56 | 21188.54 | 21530.77 | 22796.98 | 21153.93 |
| l1 3-norm BD | 1719.866 | 1543.515 | 1516.246 | 1512.4 | 1521.08 | 1656.464 | 37.45108 | 67.57934 | 66.73737 | 78.02365 | 67.95686 | 66.51245 | 7.14298 | 35.30229 | 32.07697 | 32.65884 | 32.64842 | 32.30852 | 2517.617 | 2706.617 | 2491.476 | 2519.086 | 2688.245 | 2470.969 |
| l2 3-norm BD | 1751.333 | 1510.961 | 1516.163 | 1514.81 | 1518.79 | 1658.074 | 66.28766 | 38.64332 | 66.75334 | 78.01804 | 66.94847 | 67.09068 | 32.50659 | 11.57823 | 32.25632 | 32.32539 | 32.2253 | 32.45956 | 2516.606 | 2705.533 | 2491.199 | 2518.034 | 2687.31 | 2470.926 |
| l3 3-norm BD | 1751.98 | 1544.446 | 1486.664 | 1513.27 | 1517.48 | 1658.303 | 65.47958 | 67.39749 | 39.096 | 78.03239 | 66.49712 | 67.24052 | 31.59447 | 34.64265 | 7.82387 | 32.52136 | 31.58038 | 32.51863 | 2517.107 | 2706.491 | 2489.415 | 2519.598 | 2687.107 | 2470.339 |
| l4 3-norm BD | 1751.523 | 1543.899 | 1517.328 | 1483.49 | 1522.07 | 1659.334 | 66.36948 | 67.31509 | 67.59354 | 49.69977 | 67.96119 | 67.20248 | 32.30269 | 34.81075 | 32.93691 | 7.558935 | 32.61658 | 32.28448 | 2517.248 | 2706.852 | 2491.392 | 2518.947 | 2687.751 | 2470.657 |
| l5 3-norm BD | 1753.311 | 1545.901 | 1516.143 | 1515.92 | 1488.06 | 1657.359 | 66.56918 | 67.42434 | 67.07494 | 78.55313 | 39.04714 | 66.25287 | 32.67308 | 34.67835 | 32.07084 | 32.73818 | 7.518829 | 31.55844 | 2517.357 | 2706.916 | 2491.048 | 2519.073 | 2687.064 | 2471.216 |
| l6 3-norm BD | 1751.224 | 1544.936 | 1520.698 | 1514.66 | 1519.78 | 1626.957 | 66.33501 | 67.47943 | 67.81073 | 78.43179 | 67.07613 | 38.58941 | 32.11641 | 34.73071 | 32.37629 | 32.43067 | 31.69857 | 7.459286 | 2517.511 | 2706.82 | 2490.098 | 2518.297 | 2687.431 | 2469.509 |
| l1 10-norm BD | 7.45E+08 | 1.06E+08 | 1.1E+08 | 9.9E+07 | 9.1E+07 | 1.65E+08 | 773.3514 | 831.1445 | 900.464 | 1718.299 | 846.4625 | 864.5718 | 0.293932 | 1.233024 | 0.782131 | 0.615488 | 0.748684 | 0.707943 | 0.003545 | 0.370181 | 0.403135 | 0.28582 | 0.421482 | 0.408148 |
| l2 10-norm BD | 7.45E+08 | 1.06E+08 | 1.1E+08 | 9.9E+07 | 9.1E+07 | 1.65E+08 | 773.7523 | 830.5577 | 900.2781 | 1718.625 | 846.2303 | 864.6849 | 0.61333 | 0.860265 | 0.732687 | 0.725046 | 0.727329 | 0.727967 | 0.330493 | 0.003207 | 0.368537 | 0.382603 | 0.415105 | 0.422372 |
| l3 10-norm BD | 7.45E+08 | 1.06E+08 | 1.1E+08 | 9.9E+07 | 9.1E+07 | 1.65E+08 | 773.8534 | 831.2133 | 900.141 | 1718.575 | 846.1995 | 864.7488 | 0.63865 | 1.196859 | 0.410611 | 0.735479 | 0.634941 | 0.718673 | 0.347069 | 0.317821 | 0.00318 | 0.406619 | 0.311682 | 0.405827 |
| l4 10-norm BD | 7.45E+08 | 1.06E+08 | 1.1E+08 | 9.9E+07 | 9.1E+07 | 1.65E+08 | 773.8442 | 831.1647 | 900.4524 | 1718.06 | 846.5443 | 864.7191 | 0.585811 | 1.241436 | 0.824513 | 0.351863 | 0.819103 | 0.694199 | 0.281852 | 0.379772 | 0.457535 | 0.002963 | 0.49113 | 0.376261 |
| l5 10-norm BD | 7.45E+08 | 1.06E+08 | 1.1E+08 | 9.9E+07 | 9.1E+07 | 1.65E+08 | 773.8727 | 831.1471 | 900.3779 | 1718.562 | 845.8668 | 864.717 | 0.691508 | 1.273044 | 0.735977 | 0.814864 | 0.342117 | 0.783273 | 0.40501 | 0.417533 | 0.353985 | 0.483609 | 0.003094 | 0.479598 |
| l6 10-norm BD | 7.45E+08 | 1.06E+08 | 1.1E+08 | 9.9E+07 | 9.1E+07 | 1.65E+08 | 773.9967 | 831.1642 | 900.7065 | 1718.531 | 846.6509 | 864.2966 | 0.703456 | 1.22497 | 0.82352 | 0.679747 | 0.840384 | 0.33448 | 0.403348 | 0.352543 | 0.421798 | 0.330755 | 0.503257 | 0.002948 |

Figure 8: Different Bregman divergences between all the final points and all the initial points for different mirrors in MNIST dataset using a standard CNN. Note that the smallest element in every single row is on the diagonal, which confirms the theoretical results.

## B.2 CIFAR-10 Experiments

### B.2.1 Closest Minimum for Different Mirror Descents with Fixed Initialization

We provide the distances from final points (global minima) obtained by different algorithms from the same initialization, measured in different Bregman divergences for CIFAR-10 classification task using ResNet-18. Note that in all tables the smallest element in each row is on the diagonal, which means the point achieved by each mirror has the smallest Bregman divergence to the initialization corresponding to that mirror, among all mirrors. Tables 13, 14, 15, 16, 17, 18, 19, 20 depict these results for 8 different initializations. The rows are the distance metrics used as the Bregman Divergences with specified potentials. The columns are the global minima obtained using specified SMD algorithms.

Table 13: CIFAR-10 Initial Point 1

|  | SMD 1-norm | SMD 2-norm (SGD) | SMD 3-norm | SMD 10-norm |
|---|---|---|---|---|
| 1-norm BD | 189 | $9.58 \times 10^3$ | $4.19 \times 10^4$ | $2.34 \times 10^5$ |
| 2-norm BD | $3.12 \times 10^3$ | 597 | $1.28 \times 10^3$ | $6.92 \times 10^3$ |
| 3-norm BD | $4.31 \times 10^4$ | 119 | 55.8 | $1.87 \times 10^2$ |
| 10-norm BD | $1.35 \times 10^{14}$ | 869 | $6.34 \times 10^{-5}$ | $2.64 \times 10^{-8}$ |

Table 14: CIFAR-10 Initial Point 2

|  | SMD 1-norm | SMD 2-norm (SGD) | SMD 3-norm | SMD 10-norm |
|---|---|---|---|---|
| 1-norm BD | 275 | $9.86 \times 10^3$ | $4.09 \times 10^4$ | $2.38 \times 10^5$ |
| 2-norm BD | $4.89 \times 10^3$ | 607 | $1.23 \times 10^3$ | $7.03 \times 10^3$ |
| 3-norm BD | $9.21 \times 10^4$ | 104 | 53.5 | $1.88 \times 10^2$ |
| 10-norm BD | $1.17 \times 10^{15}$ | 225 | 0.000102 | $2.65 \times 10^{-8}$ |

Table 15: CIFAR-10 Initial Point 3

|  | SMD 1-norm | SMD 2-norm (SGD) | SMD 3-norm | SMD 10-norm |
|---|---|---|---|---|
| 1-norm BD | 141 | $9.19 \times 10^3$ | $4.1 \times 10^4$ | $2.34 \times 10^5$ |
| 2-norm BD | $3.15 \times 10^3$ | 562 | $1.24 \times 10^3$ | $6.89 \times 10^3$ |
| 3-norm BD | $4.31 \times 10^4$ | 107 | 53.5 | $1.85 \times 10^2$ |
| 10-norm BD | $6.83 \times 10^{13}$ | 972 | $7.91 \times 10^{-5}$ | $2.72 \times 10^{-8}$ |

Table 16: CIFAR-10 Initial Point 4

|  | SMD 1-norm | SMD 2-norm (SGD) | SMD 3-norm | SMD 10-norm |
|---|---|---|---|---|
| 1-norm BD | 255 | $9.77 \times 10^3$ | $4.18 \times 10^4$ | $2.36 \times 10^5$ |
| 2-norm BD | $3.64 \times 10^3$ | 594 | $1.26 \times 10^3$ | $6.96 \times 10^3$ |
| 3-norm BD | $5.5 \times 10^4$ | 116 | 54 | $1.87 \times 10^2$ |
| 10-norm BD | $3.74 \times 10^{14}$ | 640 | $5.33 \times 10^{-5}$ | $2.67 \times 10^{-8}$ |

Table 17: CIFAR-10 Initial Point 5

|  | SMD 1-norm | SMD 2-norm (SGD) | SMD 3-norm | SMD 10-norm |
|---|---|---|---|---|
| 1-norm BD | 113 | $9.48 \times 10^3$ | $4.15 \times 10^4$ | $2.32 \times 10^5$ |
| 2-norm BD | $2.95 \times 10^3$ | 572 | $1.27 \times 10^3$ | $6.85 \times 10^3$ |
| 3-norm BD | $3.68 \times 10^4$ | 109 | 56.2 | $1.84 \times 10^2$ |
| 10-norm BD | $2.97 \times 10^{13}$ | 151 | $5.74 \times 10^{-5}$ | $2.61 \times 10^{-8}$ |

Table 18: CIFAR-10 Initial Point 6

|  | SMD 1-norm | SMD 2-norm (SGD) | SMD 3-norm | SMD 10-norm |
|---|---|---|---|---|
| 1-norm BD | 128 | $9.25 \times 10^3$ | $4.25 \times 10^4$ | $2.34 \times 10^5$ |
| 2-norm BD | $2.71 \times 10^3$ | 558 | $1.29 \times 10^3$ | $6.89 \times 10^3$ |
| 3-norm BD | $3.34 \times 10^4$ | 104 | 55.3 | $1.85 \times 10^2$ |
| 10-norm BD | $2.61 \times 10^{13}$ | 612 | $4.74 \times 10^{-5}$ | $2.62 \times 10^{-8}$ |

Table 19: CIFAR-10 Initial Point 7

|  | SMD 1-norm | SMD 2-norm (SGD) | SMD 3-norm | SMD 10-norm |
|---|---|---|---|---|
| 1-norm BD | 223 | $9.76 \times 10^3$ | $4.38 \times 10^4$ | $2.27 \times 10^5$ |
| 2-norm BD | $2.41 \times 10^3$ | 599 | $1.37 \times 10^3$ | $6.65 \times 10^3$ |
| 3-norm BD | $2.3 \times 10^4$ | 116 | 61 | $1.78 \times 10^2$ |
| 10-norm BD | $4.22 \times 10^{12}$ | 679 | $6.42 \times 10^{-5}$ | $2.55 \times 10^{-8}$ |

Table 20: CIFAR-10 Initial Point 8

|  | SMD 1-norm | SMD 2-norm (SGD) | SMD 3-norm | SMD 10-norm |
|---|---|---|---|---|
| 1-norm BD | 145 | $9.37 \times 10^3$ | $4.17 \times 10^4$ | $2.36 \times 10^5$ |
| 2-norm BD | $2.48 \times 10^3$ | 576 | $1.26 \times 10^3$ | $6.99 \times 10^3$ |
| 3-norm BD | $2.85 \times 10^4$ | 108 | 54.5 | $1.89 \times 10^2$ |
| 10-norm BD | $1.81 \times 10^{13}$ | $1.22 \times 10^3$ | $5.2 \times 10^{-5}$ | $2.64 \times 10^{-8}$ |

### B.2.2 CLOSEST MINIMUM FOR DIFFERENT INITILIZATIONS WITH FIXED MIRROR

We provide the pairwise distances between different initial points and the final points (global minima) obtained by using fixed SMD algorithms in CIFAR-10 dataset using ResNet-18. Note that the smallest element in each row is on the diagonal, which means the closest final point to each initialization, among all the final points, is the one corresponding to that point. Tables 21, 22, 23, 24 depict these results for 4 different SMD algorithms. The rows are the initial points and the columns are the final points corresponding to each initialization.

Table 21: CIFAR-10 1-norm Bregman Divergence Between the Initial Points and the Final Points obtained by SMD 1-norm

|  | Final 1 | Final 2 | Final 3 | Final 4 | Final 5 | Final 6 | Final 7 | Final 8 |
|---|---|---|---|---|---|---|---|---|
| Initial 1 | $1.9 \times 10^2$ | $8.1 \times 10^4$ | $8.1 \times 10^4$ | $8.4 \times 10^4$ | $8 \times 10^4$ | $8.2 \times 10^4$ | $7.8 \times 10^4$ | $7.8 \times 10^4$ |
| Initial 2 | $8.1 \times 10^4$ | $2.7 \times 10^2$ | $8.1 \times 10^4$ | $8.3 \times 10^4$ | $8 \times 10^4$ | $8.2 \times 10^4$ | $7.8 \times 10^4$ | $7.9 \times 10^4$ |
| Initial 3 | $8.1 \times 10^4$ | $8.1 \times 10^4$ | $1.4 \times 10^2$ | $8.4 \times 10^4$ | $8 \times 10^4$ | $8.1 \times 10^4$ | $7.8 \times 10^4$ | $7.8 \times 10^4$ |
| Initial 4 | $8.1 \times 10^4$ | $8.1 \times 10^4$ | $8.1 \times 10^4$ | $2.5 \times 10^2$ | $8 \times 10^4$ | $8.2 \times 10^4$ | $7.8 \times 10^4$ | $7.9 \times 10^4$ |
| Initial 5 | $8.1 \times 10^4$ | $8.1 \times 10^4$ | $8.1 \times 10^4$ | $8.3 \times 10^4$ | $1.1 \times 10^2$ | $8.1 \times 10^4$ | $7.8 \times 10^4$ | $7.8 \times 10^4$ |
| Initial 6 | $8.1 \times 10^4$ | $8.1 \times 10^4$ | $8.1 \times 10^4$ | $8.4 \times 10^4$ | $8 \times 10^4$ | $1.3 \times 10^2$ | $7.8 \times 10^4$ | $7.8 \times 10^4$ |
| Initial 7 | $8.1 \times 10^4$ | $8.1 \times 10^4$ | $8.1 \times 10^4$ | $8.4 \times 10^4$ | $8 \times 10^4$ | $8.1 \times 10^4$ | $2.2 \times 10^2$ | $7.8 \times 10^4$ |
| Initial 8 | $8.1 \times 10^4$ | $8.1 \times 10^4$ | $8.1 \times 10^4$ | $8.4 \times 10^4$ | $7.9 \times 10^4$ | $8.1 \times 10^4$ | $7.8 \times 10^4$ | $1.5 \times 10^2$ |

Table 22: CIFAR-10 2-norm Bregman Divergence Between the Initial Points and the Final Points obtained by SMD 2-norm (SGD)

|  | Final 1 | Final 2 | Final 3 | Final 4 | Final 5 | Final 6 | Final 7 | Final 8 |
|---|---|---|---|---|---|---|---|---|
| Initial 1 | $6 \times 10^2$ | $2.9 \times 10^3$ | $2.8 \times 10^3$ | $2.8 \times 10^3$ | $2.8 \times 10^3$ | $2.8 \times 10^3$ | $2.8 \times 10^3$ | $2.8 \times 10^3$ |
| Initial 2 | $2.8 \times 10^3$ | $6.1 \times 10^2$ | $2.8 \times 10^3$ | $2.8 \times 10^3$ | $2.8 \times 10^3$ | $2.8 \times 10^3$ | $2.8 \times 10^3$ | $2.8 \times 10^3$ |
| Initial 3 | $2.8 \times 10^3$ | $2.9 \times 10^3$ | $5.6 \times 10^2$ | $2.8 \times 10^3$ | $2.8 \times 10^3$ | $2.8 \times 10^3$ | $2.8 \times 10^3$ | $2.8 \times 10^3$ |
| Initial 4 | $2.8 \times 10^3$ | $2.9 \times 10^3$ | $2.8 \times 10^3$ | $5.9 \times 10^2$ | $2.8 \times 10^3$ | $2.8 \times 10^3$ | $2.8 \times 10^3$ | $2.8 \times 10^3$ |
| Initial 5 | $2.8 \times 10^3$ | $2.9 \times 10^3$ | $2.8 \times 10^3$ | $2.8 \times 10^3$ | $5.7 \times 10^2$ | $2.8 \times 10^3$ | $2.8 \times 10^3$ | $2.8 \times 10^3$ |
| Initial 6 | $2.8 \times 10^3$ | $2.9 \times 10^3$ | $2.8 \times 10^3$ | $2.8 \times 10^3$ | $2.8 \times 10^3$ | $5.6 \times 10^2$ | $2.8 \times 10^3$ | $2.8 \times 10^3$ |
| Initial 7 | $2.8 \times 10^3$ | $2.9 \times 10^3$ | $2.8 \times 10^3$ | $2.8 \times 10^3$ | $2.8 \times 10^3$ | $2.8 \times 10^3$ | $6 \times 10^2$ | $2.8 \times 10^3$ |
| Initial 8 | $2.8 \times 10^3$ | $2.9 \times 10^3$ | $2.8 \times 10^3$ | $2.8 \times 10^3$ | $2.8 \times 10^3$ | $2.8 \times 10^3$ | $2.8 \times 10^3$ | $5.8 \times 10^2$ |

Table 23: CIFAR-10 3-norm Bregman Divergence Between the Initial Points and the Final Points obtained by SMD 3-norm

|  | Final 1 | Final 2 | Final 3 | Final 4 | Final 5 | Final 6 | Final 7 | Final 8 |
|---|---|---|---|---|---|---|---|---|
| Initial 1 | 55.844 | 103.47 | 103.61 | 104.05 | 106.2 | 105.32 | 110.88 | 104.56 |
| Initial 2 | 105.87 | 53.455 | 103.68 | 104.04 | 106.31 | 105.34 | 110.93 | 104.58 |
| Initial 3 | 105.89 | 103.59 | 53.527 | 104.09 | 106.29 | 105.35 | 110.99 | 104.55 |
| Initial 4 | 105.83 | 103.54 | 103.64 | 53.978 | 106.23 | 105.3 | 110.87 | 104.54 |
| Initial 5 | 105.82 | 103.55 | 103.64 | 104 | 56.161 | 105.34 | 110.88 | 104.55 |
| Initial 6 | 105.91 | 103.6 | 103.66 | 104.1 | 106.28 | 55.316 | 110.94 | 104.55 |
| Initial 7 | 105.87 | 103.51 | 103.67 | 103.98 | 106.26 | 105.25 | 61.045 | 104.5 |
| Initial 8 | 105.77 | 103.54 | 103.59 | 104.04 | 106.25 | 105.28 | 110.88 | 54.509 |

Table 24: CIFAR-10 10-norm Bregman Divergence Between the Initial Points and the Final Points obtained by SMD 10-norm

| | Final 1 | Final 2 | Final 3 | Final 4 | Final 5 | Final 6 | Final 7 | Final 8 |
|---|---|---|---|---|---|---|---|---|
| Initial 1 | $2.64 \times 10^{-8}$ | $2.89 \times 10^{-8}$ | $2.99 \times 10^{-8}$ | $2.81 \times 10^{-8}$ | $2.85 \times 10^{-8}$ | $2.82 \times 10^{-8}$ | $2.66 \times 10^{-8}$ | $2.82 \times 10^{-8}$ |
| Initial 2 | $2.79 \times 10^{-8}$ | $2.65 \times 10^{-8}$ | $2.83 \times 10^{-8}$ | $2.83 \times 10^{-8}$ | $2.71 \times 10^{-8}$ | $2.74 \times 10^{-8}$ | $2.69 \times 10^{-8}$ | $2.88 \times 10^{-8}$ |
| Initial 3 | $2.89 \times 10^{-8}$ | $2.87 \times 10^{-8}$ | $2.72 \times 10^{-8}$ | $2.94 \times 10^{-8}$ | $2.84 \times 10^{-8}$ | $2.89 \times 10^{-8}$ | $2.78 \times 10^{-8}$ | $2.94 \times 10^{-8}$ |
| Initial 4 | $2.79 \times 10^{-8}$ | $2.86 \times 10^{-8}$ | $2.92 \times 10^{-8}$ | $2.67 \times 10^{-8}$ | $2.84 \times 10^{-8}$ | $2.81 \times 10^{-8}$ | $2.69 \times 10^{-8}$ | $2.85 \times 10^{-8}$ |
| Initial 5 | $2.76 \times 10^{-8}$ | $2.88 \times 10^{-8}$ | $2.95 \times 10^{-8}$ | $2.93 \times 10^{-8}$ | $2.61 \times 10^{-8}$ | $2.73 \times 10^{-8}$ | $2.66 \times 10^{-8}$ | $2.83 \times 10^{-8}$ |
| Initial 6 | $2.80 \times 10^{-8}$ | $2.76 \times 10^{-8}$ | $2.93 \times 10^{-8}$ | $2.79 \times 10^{-8}$ | $2.76 \times 10^{-8}$ | $2.62 \times 10^{-8}$ | $2.71 \times 10^{-8}$ | $2.85 \times 10^{-8}$ |
| Initial 7 | $2.73 \times 10^{-8}$ | $2.76 \times 10^{-8}$ | $2.82 \times 10^{-8}$ | $2.79 \times 10^{-8}$ | $2.71 \times 10^{-8}$ | $2.77 \times 10^{-8}$ | $2.55 \times 10^{-8}$ | $2.83 \times 10^{-8}$ |
| Initial 8 | $2.73 \times 10^{-8}$ | $2.79 \times 10^{-8}$ | $2.85 \times 10^{-8}$ | $2.78 \times 10^{-8}$ | $2.75 \times 10^{-8}$ | $2.74 \times 10^{-8}$ | $2.73 \times 10^{-8}$ | $2.64 \times 10^{-8}$ |

### B.2.3 CLOSEST MINIMUM FOR DIFFERENT INITILIZATIONS AND DIFFERENT MIRRORS

Now we assess the pairwise distances between different initial points and final points (global minima) obtained by all different initilizations and all different mirrors (Table 8). The smallest element in each row is exactly the final point obtained by that mirror from that initialization, among all the mirrors and all the initial points.

Figure 9: Different Bregman divergences between all the final points and all the initial points for different mirrors in CIFAR-10 dataset using ResNet-18. Note that the smallest element in every single row is on the diagonal, which confirms the theoretical results.

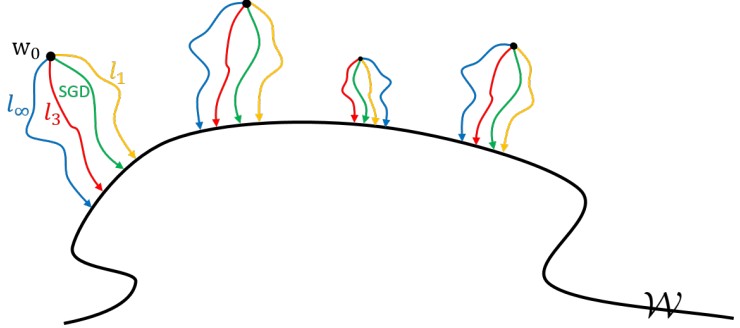

Figure 10: An illustration of the experimental results. For each initialization $w_0$, we ran different SMD algorithms until convergence to a point on the set $\mathcal{W}$ (zero training error). We then measured all the pairwise distances from different $w_\infty$ to different $w_0$, in different Bregman divergences. The closest point (among all initializations and all mirrors) to any particular initialization $w_0$ in Bregman divergence with potential $\psi(\cdot) = \|\cdot\|_q^q$ is exactly the point obtained by running SMD with potential $\|\cdot\|_q^q$ from $w_0$.

### B.3 DISTRIBUTION OF THE FINAL WEIGHTS OF THE NETWORK

One may be curious to see how the final weights obtained by these different mirrors look like, and whether, for example, mirror descent corresponding to the $\ell_1$-norm potential induces sparsity. We examine the distribution of the weights in the network for these algorithms starting from the same initialization. Fig. 11 shows the histogram of the initial weights, which follows a half-normal distribution. Figs. 12 (a), (b), (c), (d) show the histogram of the weights for $\ell_1$-SMD, $\ell_2$-SMD (SGD), $\ell_3$-SMD, and $\ell_{10}$-SMD, respectively. Note that each of the four histograms corresponds to an $11 \times 10^6$-dimensional weight vector that perfectly interpolates the data. Even though, perhaps as expected, the weights remain quite small, the histograms are drastically different. The histogram of the $\ell_1$-SMD has more weights at and close to zero, which again confirms that it induces sparsity. However, as will be shown in the next section, this is not necessarily good for generalization (in fact, it turns out that $\ell_{10}$-SMD has a much better generalization). The histogram of the $\ell_2$-SMD (SGD) looks almost identical to the histogram of the initialization, whereas the $\ell_3$ and $\ell_{10}$ histograms are shifted to the right, so much so that almost all weights in the $\ell_{10}$ solution are non-zero and in the range of 0.005 to 0.04. For comparison, all the distributions are shown together in Fig. 12(e).

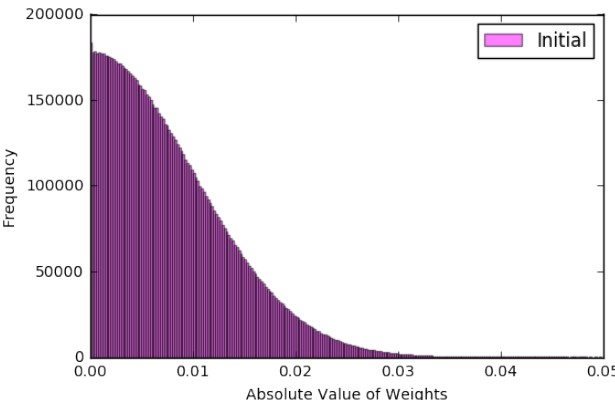

Figure 11: Histogram of the absolute value of the initial weights in the network (half-normal distribution)

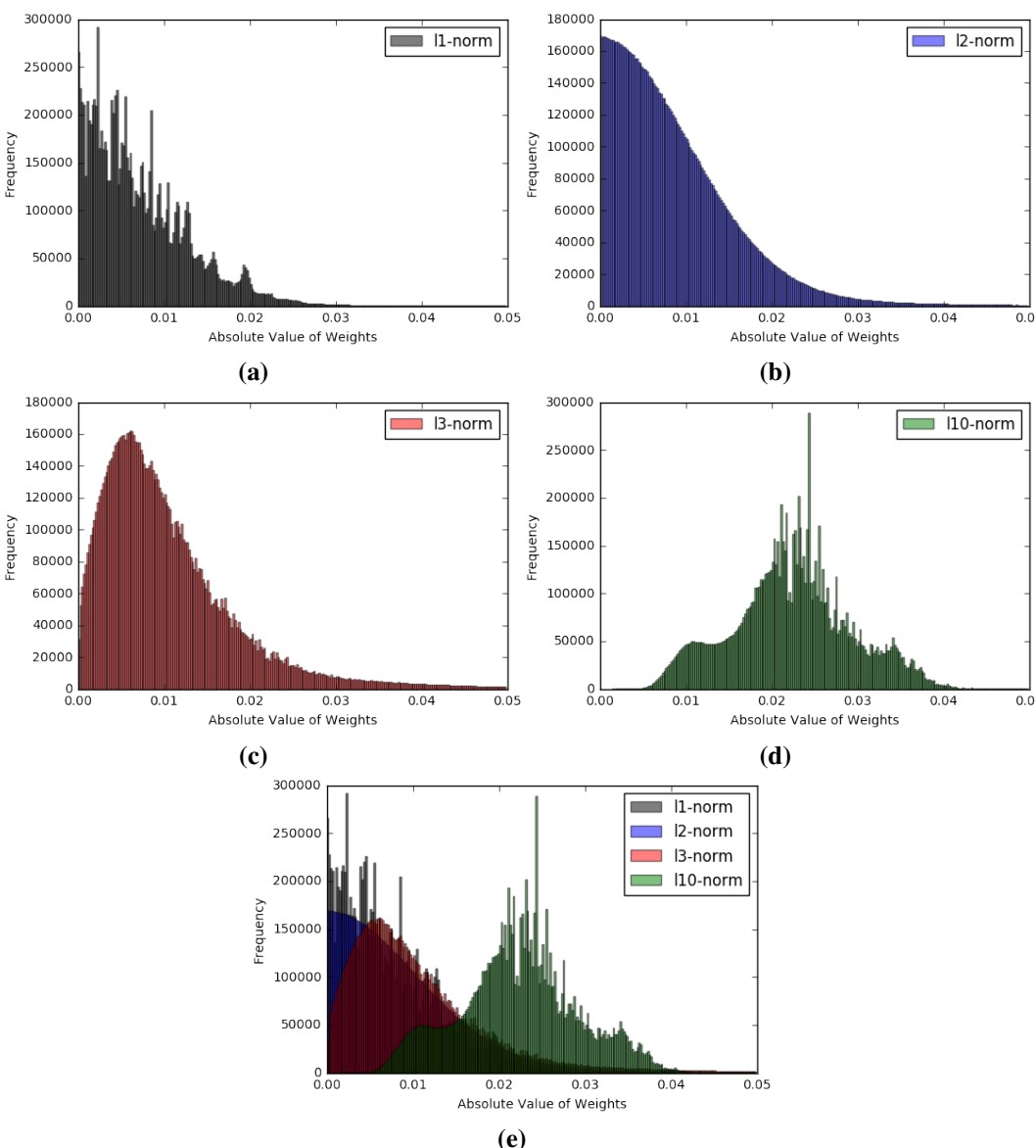

Figure 12: Histogram of the absolute value of the final weights in the network for different SMD algorithms: (a) $\ell_1$-SMD, (b) $\ell_2$-SMD (SGD), (c) $\ell_3$-SMD, (d) $\ell_{10}$-SMD. Note that each of the four histograms corresponds to an $11 \times 10^6$-dimensional weight vector that perfectly interpolates the data. Even though the weights remain quite small, the histograms are drastically different. $\ell_1$-SMD induces sparsity on the weights, as expected. SGD does not seem to change the distribution of the weights significantly. $\ell_3$-SMD starts to reduce the sparsity, and $\ell_{10}$ shifts the distribution of the weights significantly, so much so that almost all the weights are non-zero.

### B.4 GENERALIZATION ERRORS OF DIFFERENT MIRRORS/REGULARIZERS

In this section, we compare the performance of the SMD algorithms discussed before on the test set. This is important for understanding the effect of different regularizers on the generalization of deep networks.

For MNIST, perhaps not surprisingly, all the four SMD algorithms achieve around $99\%$ or higher accuracy. For CIFAR-10, however, there is a significant difference between the test errors of different mirrors/regularizers on the same architecture. Fig. 13 shows the test accuracies of different algorithms with eight random initializations around zero, as discussed before. Counter-intuitively, $\ell_{10}$ performs consistently best, while $\ell_1$ performs consistently worse. We should reiterate that the loss function is exactly the same in all these experiments, and all of them have been trained to fit the training set perfectly (zero training error). Therefore, the difference in generalization errors is purely the effect of implicit regularization by different algorithms. This result suggests the importance of a comprehensive study of the role of regularization, and the choice of the best regularizer, to improve the generalization performance of deep neural networks.

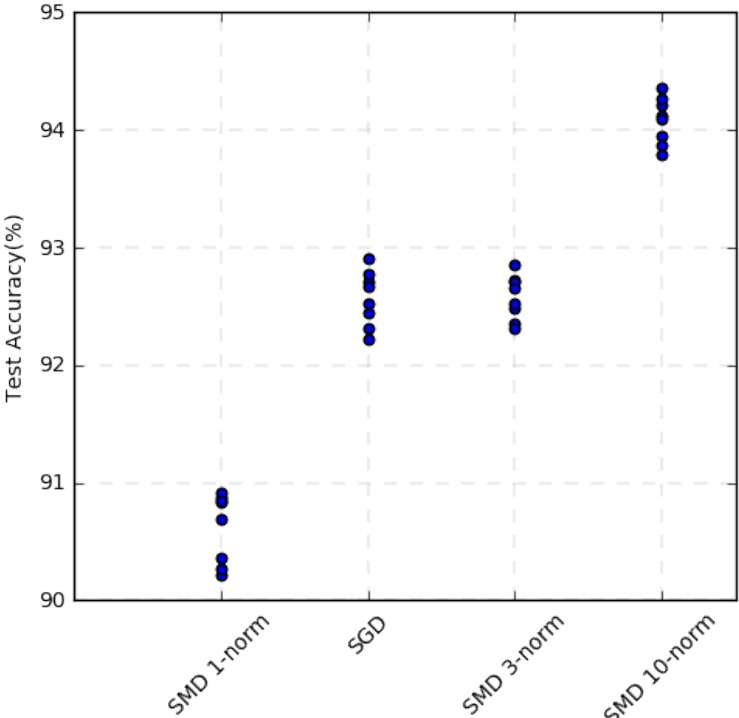

Figure 13: Generalization performance of different SMD algorithms on the CIFAR-10 dataset using ResNet-18. $\ell_{10}$ performs consistently better, while $\ell_1$ performs consistently worse.

