# OpenReview forum: "Stochastic Mirror Descent on Overparameterized Nonlinear Models"
_ICLR.cc/2020/Conference — Reject_

### Official Review · AnonReviewer1 · 2019-10-22
**Official Blind Review #1**

**Rating:** 6

**Review:**

This paper studies the optimization behaviour of stochastic mirror descent (SMD) on over-parametrized nonlinear models. It shows, rigorously and empirically, that SMD finds global minimizer that is approximately the closet global minimizer to the initial point, under the same Bregman divergence.  The paper is well written and easy to follow.

Pros:
1. The main contribution of this paper is extending the previous results on linear case into nonlinear setting, which is much more general. The results are non-trivial and interesting.
2. The empirical studies in the paper is clean and significant. It also supports the developed theory.

The main concern I have is about assumption 1 in the paper.  While in the paper the authors tend to claim that it is reasonable/mild, it is not obvious to me.
    a. Many factors are actually involved in this assumption: smoothness of \psi, smoothness of L_i and f_i, number of parameters etc.. I am not sure if it is still a high probability, when \eps needs to be small enough so that the intuition behind Figure 2 still holds.
    b. Is this assumption also made in the referred papers on the top of page 6? Could the authors develop a concrete result, given smoothness coefficients of \psi and L_i, for this assumption?
    c. Without a proper justification of assumption 1, the significance of this paper seems much weaker. For example, one can also assume the PL inequality locally true for over-parametrized function, and develop a convergence result.


**Experience Assessment:**

I have read many papers in this area.

**Review Assessment: Checking Correctness Of Derivations And Theory:**

I assessed the sensibility of the derivations and theory.

**Review Assessment: Checking Correctness Of Experiments:**

I assessed the sensibility of the experiments.

**Review Assessment: Thoroughness In Paper Reading:**

I read the paper at least twice and used my best judgement in assessing the paper.

---

> ### Author Response · Authors · 2019-11-15
> **Response to Review #1**
>
> We thank you for your kind comments and for highlighting the pros of our paper.
> In our discussion of the reasonableness/mildness of Assumption 1, we referred to the papers on the top of page 6 and the discussion of Appendix A.4. Perhaps we should have given a more explicit description of these references and brought A.4 to the main body of the paper. In Appendix A.4, the critical equations are (43), (44) and the unnumbered equation in between. Equation (43) describes the closeness of $w_0$ to $w^*$ up to quadratic terms, and depends on the matrix $M=\nabla f(x,w_0)^T\nabla f(x,w_0)$. For linear models, it is easy to argue that $\lambda_{\min}(M)$ is of $O(p)$. For nonlinear models, provided the network is “wide enough,” it can also be shown that $\lambda_{\min}(M)$ is of $O(p)$. Equation (44) concludes that the distance between $w_0$ and $w^*$ is of $O(n/p)$. For a fixed $n$ (number of data points), $w_0$ can get arbitrarily close to $w^*$ as $p$ (the size of the network) grows. The $\epsilon$ in Assumption 1 is essentially of order $n/p$. This is why we say that Assumption 1 is reasonable in the highly overparameterized regime, since here $p\gg n$. For example, for the CIFAR-10 dataset  $n=50,000$ and $p=11\times 10^6$, resulting in $n/p\approx 0.005$.
>
> a) As the reviewer has correctly noted, the arguments above, as well as those used in the papers on the top of page 6, rely on the smoothness of $L_i(\cdot)$, $f_i(\cdot)$, and, for SMD, of $\psi(\cdot)$. The differentiability of $\psi$ is part of the definition of mirror descent, and the differentiability of $L_i$ and $f_i$ are assumed for the existence of the gradients.
>
> With regards to whether Assumption 1 holds with high probability with the choice of $w_0$, we should remark that Figure 2 is misleading since it is a 1-dimensional representation. In Figure 2, it appears that if $w_0$ is moved sufficiently to the left or right, we are no longer within $\epsilon$ of a minimizer of $L_i(\cdot)$. However, in the highly overparameterized setting, the set of global minima is a very large set (a $p-n$-dimensional set in a $p$ dimensional space, where $p\gg n$). As argued earlier, no matter how $w_0$ is moved, it will remain within distance $O(n/p)$ of some global minimum. It is in this sense that Assumption 1 holds with high probability: when $p\gg n$, any arbitrarily chosen $w_0$, whp will be with distance $O(n/p)$ of the set of global minima.
>
> b) The papers mentioned on the top of page 6 make deductions on the closeness that are similar to our Assumption 1. Their deductions rely on the high overparameterization ($p\gg n$), the smoothness of $L_i(\cdot)$ and $f_i(\cdot)$, as well as the wideness of the network.
> It is possible to obtain concrete bounds relating $\epsilon$ to $p$, $n$, the width of the network, as well as to smoothness properties of $\psi(\cdot)$, $L_i(\cdot)$, and $f_i(\cdot)$; however, it would require an effort beyond the current revision.
>
> c) We further remark that, as correctly noted by the reviewer, one might also be able to use PL condition to prove similar results. In fact, (Oymak & Soltanolkotabi 2019) do that in their Section 5 for the case of GD. But proving it for SGD as well as SMD would be much more challenging. Our current results hold for SGD and SMD.
>
> We hope the above discussions address your main concern.

---

> > ### Comment · AnonReviewer1 · 2019-11-15
> > **After Rebuttal**
> >
> > Thanks for the response.
> >
> > I can understand that it is easy to find w0 that is close to the manifold of W. However, to achieve that, f needs to be a large network. As f becomes larger, the smoothness of f may reduce significantly, and thus the smoothness of L_i. How do I know that D_{L_i}(w, w') >0 can still hold with high probability?

---

> > > ### Author Response · Authors · 2019-11-15
> > > **Response**
> > >
> > > We thank the reviewer for their thoughtful comment. If indeed, as the size of the network grows, the smoothness of $L_i$ and $f$ around the interpolating set of weights were to significantly decrease to the point of being rapidly oscillatory, then that *could* be a problem for our argument that $D_{L_i}(w, w') >0$ holds with high probability. Since we do do not know whether this is the case or not, to the best of our abilities in the time available for this response, we surveyed the relevant literature. We were unable to find any "empirical" evidence that suggested this (we tried hard). On the theoretical side, we found a few papers that upper bound the Lipschitz constant of neural networks (in worst case, it is known that it can grow exponentially with the depth of the network). However, these are worst-case analyses and it is not clear whether they hold typically in practice. (They require that the directions of non-smoothness perfectly align in adjacent layers of the network, which is unlikely.)
> > >
> > > As a result, we are not certain whether the issue raised by the reviewer occurs in practice. The empirical evidence of our paper seems to suggest otherwise, since the theory is strongly confirmed. We will add a discussion of this issue in the final version of the paper (should it be accepted).

---

### Official Review · AnonReviewer3 · 2019-10-23
**Official Blind Review #3**

**Rating:** 3

**Review:**

In this paper, the authors study the behavior of stochastic mirror descent on overparameterized nonlinear models. Especially, the authors show that for appropriate initialization, SMO converges to a global minimum with a minimal distance to the initialization. The authors also report some experimental results to verify this theory.

The implicit regularization of SGD/SMD is widely studied in the literature. Although it is claimed that the paper improved existing results by either considering nonlinear models or identify more precise implication regularization, the contribution is a bit minor. In particular, the fundamental identity in Lemma 6 is exactly a specific case of Lemma 4 in  Azizan & Hassibi (2018) with no noises. Moreover, Azizan & Hassibi (2018) also discussed the implicit regularization of SMD in their Theorem 10 for nonlinear models with some localization arguments. This paper considers similar localization arguments with some different assumptions.

After rebuttal:
I am sorry to select by mistake a N/A for the correctness of experiments.
I have read the authors' rebuttal and would not like to change my score. The fundamental identity in Lemma 6 was already derived in Azizan & Hassibi (2018). This paper identifies some assumptions to show the implicit regularization studied in Azizan & Hassibi (2018) with heuristic argument. However, these assumptions are very strong and with the assumptions the argument follows similarly to those in Azizan & Hassibi (2018). Indeed, as stated in Azizan & Hassibi (2018), the convergence in linear case depends on the non-negativity of $D_{L_i}(w,w_{i-1})$. In this paper, the authors just assume that this non-negativity holds locally in Assumption 1. In my opinion, this seems not a rigorous way to formulate assumptions.

**Experience Assessment:**

I have published one or two papers in this area.

**Review Assessment: Checking Correctness Of Derivations And Theory:**

I assessed the sensibility of the derivations and theory.

**Review Assessment: Checking Correctness Of Experiments:**

I did not assess the experiments.

**Review Assessment: Thoroughness In Paper Reading:**

I read the paper at least twice and used my best judgement in assessing the paper.

---

> ### Author Response · Authors · 2019-11-15
> **Response to Review #3**
>
> We thank you for your comments.
> We agree that the implicit regularization of SGD/SMD has been studied in the recent literature. However, we would like to emphasize the following points:
>
> 1. All the results for SMD in the literature (except for [Azizan & Hassibi, 2019], discussed in item 2 below) are for *linear* models.
>
> 2. The nonlinear result discussed in Section 5.2 of [Azizan & Hassibi, 2019] is quite informal. The argument for Theorem 10 in that paper is mostly heuristic and the statement of the theorem itself is also somewhat informal. In the current paper, we have made the statement for the nonlinear case precise: We separately identify the conditions required for convergence (Assumption 1 in Theorem 3) and implicit regularization (Assumption 2 in Theorem 4). We explicitly give a bound on the step size (requiring the strict convexity of $\psi(\cdot)-\eta L_i(\cdot)$ for all $i$, which generalizes the bound for SMD in linear models). We make the notions of “closeness” and “locality” precise in terms of the corresponding Bregman divergence (not in terms of the imprecise notion of distance in Theorem 10 of [AH ‘19]). And, finally, we have two statements 1 and 2 in Theorem 4, rather than the single statement 2 in Theorem 10 of [AH ‘19].
> While the reviewer contends that those improvements are a bit minor, we beg to differ. Establishing Theorems 3 and 4 took a significant effort and multiple attempts. However, irrespective of the effort, we believe there is value in rigorously stating and proving the conditions and manner of convergence and implicit regularization for SMD on overparameterized nonlinear models.
> We also consider our experimental results to be a significant contribution of the paper, as detailed next.
>
> 3. We should emphasize that none of the papers studying the implicit regularization of SGD/SMD, and in particular [AH ‘19], provide any experimental results. In fact, to the best of our knowledge, the current paper provides the first experimental results on applying SMD to deep learning. We provide extensive and systematic experiments on real datasets (both MNIST and CIFAR-10) using standard off-the-shelf architectures (ConvNet and ResNet) for various mirror descent algorithms, various random initializations, and various Bregman divergences. All these experiments consistently confirmed the predictions of the theory, namely that among all the obtained global minima, the one we reach from a certain initialization and mirror is the one closests to the initialization in the corresponding Bregman divergence. This is noteworthy in itself. Furthermore, the histograms of Figure 6 clearly show the effect of the implicit regularization on the distribution of the interpolating weights. Finally, Figure 7 clearly shows the effect of implicit regularization on the generalization performance of the network and provides some surprising insights.
>
> In light of these facts, we would like to invite the reviewer to change their rating (as well as to change the N/A assessment for experiments).

---

### Official Review · AnonReviewer4 · 2019-11-08
**Official Blind Review #4**

**Rating:** 3

**Review:**

This paper studies the performance of the mirror gradient method when applied to the overparameterized network. The authors claim that the SMD method could find the regularized global minimize for different potential functions, in terms of minimal Bregman distance. Further experiments are carried out to back up the authors' claim.

However, the drawbacks of this paper are listed in these several points:

1) The regularization w.r.t. Bregman distance is quite different from the regularization used in network training: instead of $\|w - w_0\|$, we use $\|w\|$ more often, therefore, the virtue of sparsity shared by 1-norm is not exploited.

2) The authors' assumption in Assumption 3.1 is too fuzzy: instead of detailed analysis in the papers related to the overparameterized network, the author does not give ANY relationship between the $\epsilon$ and these three important parameters: a) network width, b) probability introduced by random initialization, c) the number of input data.

3) Therefore, according to the too strong and fuzzy assumption mentioned in the last point, Assumption 3.1 simply makes the network equals to the linear model, which leads to minor contributions according to the study of the overparameterized network.

4) The `over parameterized network' discussed now, including the work of (Li & Liang, 2018; Du et al., 2018; Azizan & Hassibi, 2019; Allen-Zhu et al., 2019; Cao & Gu, 2019), are mainly focused on the `network of infinite width', however, in the authors' experiment, the network architecture, e.g. ResNet18, is more to be a `very deep network' rather than a `very wide network'. Therefore, the authors' theory is built on Assumption 3.1, which is based on `network of infinite width', while the experiment is built on the `very deep network'. The result of the experiments is not enough to support the authors' theorem.

In conclusion, I am convinced that the authors' work is over-claimed in this paper.

**Experience Assessment:**

I have published one or two papers in this area.

**Review Assessment: Checking Correctness Of Derivations And Theory:**

I assessed the sensibility of the derivations and theory.

**Review Assessment: Checking Correctness Of Experiments:**

I assessed the sensibility of the experiments.

**Review Assessment: Thoroughness In Paper Reading:**

I read the paper thoroughly.

---

> ### Author Response · Authors · 2019-11-15
> **Response to Review #4**
>
> We thank the reviewer for spending the time to review our paper and provide a review.
>
> Having said that, we feel it would be remiss not to make a few comments before addressing the reviewer’s concerns. The authors of the paper have been in the business of writing/reviewing/revising papers for a considerable amount of time. As a result, we find it difficult to fathom how any reviewer in good faith could give the lowest possible rating (a rating of 1 here) for a paper that the other reviewers consider well-written, for which no glaring (or even slight) technical faults have been found, and for which no clear flaw in the empirical sections have been discovered. This is a paper that we put a great deal of effort into: both into the exposition, as well as the considerable work we put into deriving the technical results, and also the care we took in creating and implementing experiments and objectively interpreting the results. The fact that all this can be easily dismissed with a score of 1 is quite startling. We believe that such arbitrary scoring will just discourage authors from submitting to ICLR, and will detract from the high standards the conference aspires to adhere to.
>
> Understanding how the reviewer reached a rating of 1 is very difficult for us. However, responding to his/her critiques is quite easy.
>
> 1) It is true that our general result states that SMD can regularize for $D_{\psi}(w,w_0)$, which the reviewer characterizes as $\|w-w_0\|$. However, as we make it clear in Corollaries 2 and 5, in paragraph 5 of page 2, and elsewhere in the paper, one can regularize for $\psi(w)$ by choosing $w_0$ to be the minimizer of $\psi(\cdot)$. For example, we can regularize for any $\ell_p$ norm by initializing around zero. In fact, this is what has been done in our experiments on MNIST and CIFAR-10, where in the first paragraph of Section 4.1 we say “We randomly initialize the parameters of the networks around zero.” Therefore all our experiments do regularize for the norms we claim. In particular, we regularize for $\|w\|_1$ when we seek to encourage sparsity; the fact that the resulting $w$ is sparser than the $w$ found for other regularizers is clear from Figure 6.
>
> So we do not see how our regularization is “quite different from the regularization used in network training”. The reviewer is mistaken.
>
> 2) We refer the reviewer to our response to Reviewer 1. The $\epsilon$ is on the order of $O(n/p)$, where $n$ is the number of data points and $p$ is the number of parameters.
>
> 3) In view of the discussion on Assumption 1 and Theorems 3 and 4, our result is *not* equivalent to the linear case. It does require $p\gg n$, large width, smoothness, random inputs, etc, but it is not equivalent to the linear case. The proof of the nonlinear result required considerable effort. Furthermore, in every single one of our experiments (which are certainly nonlinear), the result was consistent with the predictions of the theory.
>
> 4) Neither our Assumption 1 nor the results in the mentioned references require "infinite width." As long as the network is sufficiently wide such that $\lambda_{\min}(\nabla f(x,w_0)^T\nabla f(x,w_0))=O(p)$, the result holds true, and this is reasonable for typical networks such as ResNet-18. The reviewer's subjective characterization of ResNet-18 as "a very deep network" is certainly incorrect (in fact, there are much deeper networks such as ResNet-50 and ResNet-101, with 50 and 101 layers respectively).

---

### Decision · Program_Chairs · 2019-12-19

**Decision:**

Reject

**Comment:**

This paper takes results related to the convergence and implicit regularization of stochastic mirror descent, as previously applied within overparameterized linear models, and extends them to the nonlinear case.  Among other things, conditions are derived for guaranteeing convergence to a global minimizer that is (nearly) closest to the initialization with respect to a divergence that depends upon the mirror potential.  Overall the paper is well-written and likely at least somewhat accessible even for non-experts in this field.

That being said, two reviewers voted to reject while one chose accept; however, during the rebuttal period the accept reviewer expressed a somewhat borderline sentiment.  As for the reviewers that voted to reject, a common criticism was the perceived similarity with reference (Azizan and Hassibi, 2019), as well as unsettled concerns about the reasonableness of the assumptions involved (e.g., Assumption 1).  With respect to the former, among other similarities the proof technique from both papers relies heavily on Lemma 6.  It was then felt that this undercut the novelty somewhat.

Beyond this though, even the accept reviewer raised an unsettled issue regarding the ease of finding an initialization point close to the manifold that nonetheless satisfies the conditions of Assumption 1.  In other words, as networks become more complex such that points are closer to the manifold of optimal solutions, further non-convexity could be introduced such that the non-negativity of the stated divergence becomes more difficult to achieve.  While the author response to this point is reasonable, it feels a bit like thoughtful speculation forged in the crunch time of a short rebuttal period, and possibly subject to change upon further reflection.  In this regard a less time-constrained revision could be beneficial (including updates to address the other points mentioned above), and I am confident that this work can be positively received at another venue in the near future.